# Value-Guided Search for Efficient Chain-of-Thought Reasoning

**Kaiwen Wang**[*1], **Jin Peng Zhou**[*1], **Jonathan Chang**[*4],
**Zhaolin Gao**[1], **Nathan Kallus**[1,3], **Kianté Brantley**[2], and **Wen Sun**[1]

[1]Cornell University    [2]Harvard University    [3]Netflix    [4]Databricks

## Abstract

In this paper, we propose a simple and efficient method for value model training on long-context reasoning traces. Compared to existing process reward models (PRMs), our method does not require a fine-grained notion of "step," which is difficult to define for long-context reasoning models. By collecting a dataset of 2.5 million reasoning traces, we train a 1.5B token-level value model and apply it to DeepSeek models for improved performance with test-time compute scaling. We find that block-wise value-guided search (`VGS`) with a final weighted majority vote achieves better test-time scaling than standard methods such as majority voting or best-of-$n$. Moreover, `VGS` significantly reduces the inference FLOPs required to achieve the same performance of majority voting. Our dataset, model and codebase are open-sourced at `https://github.com/kaiwenw/value-guided-search`.

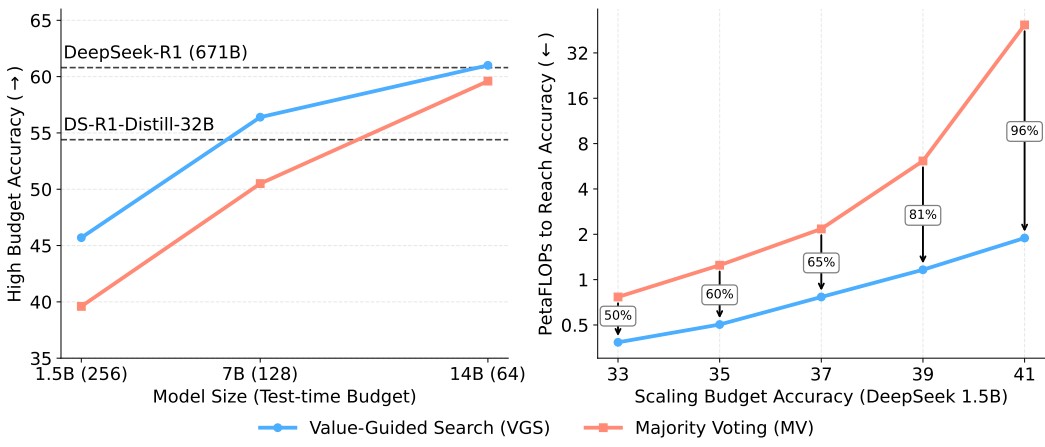

Figure 1: **Performance and Efficiency of Value Guidance:** (Left) Value-guided search improves the overall quality of `DeepSeek-R1-Distill` responses across combined competition math benchmarks (AIME 24, 25 & HMMT Feb 24, 25). The inference budget for 1.5B, 7B and 14B are 256, 128 and 64 generations, respectively. (Right) Value-guided search also reduces the inference FLOPs required to achieve the same accuracy levels as majority voting, a standard TTC scaling baseline, showing value-guidance is promising for improving efficiency.

---

[*]Equal contribution. Correspondence to `{kw437,jz563}@cornell.edu`.

39th Conference on Neural Information Processing Systems (NeurIPS 2025).

# 1 Introduction

Recent large language models (LLMs), such as OpenAI o1 & o3, Claude Sonnet 3.7, Gemini Pro 2.5 and DeepSeek R1 [19] are trained via reinforcement learning (RL) to "think" for many tokens before generating a final answer. Through multi-step reasoning and self-correction, these *reasoning models* have state-of-the-art performance in competition math, coding [16] and scientific research [38], often surpassing the average human. However, this enhanced capability comes at a cost: each generation involves a long chain-of-thought (CoT), thus requiring more inference compute. Further, these CoT traces can often be repetitive and get stuck in unproductive loops [31]. This raises two questions. Can we extract the same performance at a fraction of the inference compute by refining the thinking process? Can we improve the performance ceiling of these models with productive search methods?

Search with guidance models is a natural solution that addresses longer chain-of-thought reasoning by managing the exponential growth of possible paths with guidance models identifying optimal routes [15, 40, 36]. Prior works that combined search with LLMs proposed to guide search with process reward models (PRMs), predicting the correctness of each step (*e.g.*, delimited by newlines) in the model-generated solution [24, 45, 51]. While PRM-guided search has been shown to improve test-time compute (TTC) [6, 37, 35, 26], it is challenging to scale existing PRM training techniques to long-context reasoning models. First, existing methods require a pre-defined notion of "step," but, per Guo et al. [19], "it is challenging to explicitly define a fine-grain step in general reasoning." Second, even if we can define a "step," collecting step-wise labels is prohibitively expensive, since it requires annotations from humans [24], LLM-as-a-Judge [51], or multiple Monte Carlo roll-outs [45, 28]. Thus, there has been limited success to scale PRMs to long-context reasoning models [19].

We propose value-guided search (VGS) – a block-level search method guided by a token-level value model – as a promising approach to scale TTC for reasoning models. In Section 2, we present an effective pipeline for value model training on tasks with outcome labels, such as competition math. Our data pipeline collects solution prefixes from various models and then, starting from random prefixes, generates completed solutions using a *lean reasoning model* (*e.g.*, `DeepSeek-R1-Distill-1.5B`). Notably, our data collection does not require a pre-defined notion of step and is more efficient than existing techniques [45, 28]. With this pipeline, we collect a dataset of 2.5 million math reasoning traces (over 30 billion tokens) from a filtered subset of the OpenR1-Math dataset [2]. Then, we train a 1.5B token-level value model called `DeepSeek-VM-1.5B` by regressing (via classification) the final reward of the completed solution.

Next, in Section 3, we apply our value model to perform block-wise search with DeepSeek models [19] on competition math, where we evaluate on four prestigious high school math competitions in the US (AIME 2024 & 2025 and HMMT 2024 & 2025). Our experiments show that block-wise VGS significantly improves TTC compared to majority voting or weighted majority voting, strong baselines from the literature [46, 6]. We also show that VGS with `DeepSeek-VM-1.5B` leads to higher performance than searching with state-of-the-art PRMs, demonstrating that our value model can provide better feedback. When given an inference budget of $64$ generations, VGS on `DeepSeek-R1-Distill-Qwen-14B` (total size with value model is 15.5B) is on par with `DeepSeek-R1` (671B) on our competition math evaluations (Fig. 1 left). Moreover, we show that VGS reduces the amount of inference compute required to attain the same performance as majority voting (Fig. 1 right). In summary, we find that block-wise VGS not only improves the performance ceiling of reasoning models, but also reduces the amount of inference compute required to match the performance of standard TTC methods. Our contributions are summarized below:

1. A simple recipe for token-level value model training that does not require a pre-defined notion of "step" and scales to long-context reasoning traces (Section 2).

2. Block-wise search, guided by our 1.5B value model, achieves impressive performance on four challenging math competitions, outperforming standard TTC methods (*e.g.*, best-of-$N$, majority voting) and search with existing PRMs (Section 3).

3. We open-source our dataset of 2.5 million reasoning traces, value model, and codebase (including data filtering and distributed training scripts) to support future work on applying VGS to other domains. `https://github.com/kaiwenw/value-guided-search`.

Please see Appendix A for a detailed discussion of related works.

## 2 Methods

We present an end-to-end pipeline for training a token-level value model and applying it to guide block-wise search. In Section 2.1, we introduce necessary notation and present a regression-via-classification algorithm for learning the token-level value model [14]. Then, in Section 2.2, we outline an efficient data pipeline for creating our dataset of 2.5 million reasoning traces from DeepSeek models. Finally, in Section 2.3, we describe several TTC methods and baselines, *e.g.*, best-of-$N$, (weighted) majority voting and search algorithms that can leverage our value model. While we focus on competition math in this paper, we remark that our pipeline can in principle be applied to any task with automated outcome supervision (*e.g.*, a reward model). In Appendix B, we summarize a simple recipe for applying VGS to other such domains.

### 2.1 Learning Algorithm for Value Model

We describe our training process for a language value model by performing regression via classification [7]. Let $\mathcal{V}$ be the vocabulary and let $\mathcal{S} = \bigcup_{n \in \mathbb{N}} \mathcal{V}^n$ denote the input sequence space. Given a problem prompt $x \in \mathcal{S}$ and a response $y \in \mathcal{S}$, let $\kappa = \Gamma(x, y) \in [0, 1, \ldots, K-1]$ denote its class label, where $K$ is the number of classes. Furthermore, let $r = R(x, y)$ denote the scalar reward, which we assume to be binary since we focus on competition math (see Appendix B for the general case). For our value model, $\kappa = 2$ if the response is an incomplete generation (*i.e.*, exceeds max generation length), $\kappa = 0$ if the response finished and is incorrect, and $\kappa = 1$ if the response finished and is correct. Thus, the event that $\kappa = 1$ corresponds to $r = 1$ (correct answer), and $\kappa \in \{0, 2\}$ corresponds to $r = 0$ (incorrect or exceeds max length). We adopt this convention in the rest of the paper. We remark that regression-via-classification is a standard approach that leads to better down-stream decision making than regressing via squared error [7, 21, 17, 3, 43, 44].

We employ datasets of the form $\mathcal{D} = \{(x_i, y_i, z_i, \kappa_i)\}_{i \in [N]}$, where $x_i$ is the problem prompt, $y_i$ is a partial response (which we call a "roll-in"), $z_i$ is a completion starting from $y_i$ (which we call a "roll-out"), and $\kappa_i = \Gamma(x_i, y_i, z_i)$ is the label of the full response, where $x, y, z$ denotes the concatenation of $x$, $y$ and $z$. In this paper, we assume that the completions / roll-outs $z_i$ are generated by a fixed roll-out policy $\pi^{\mathrm{ref}}$, *i.e.*, $z_i \sim \pi^{\mathrm{ref}}(\cdot \mid x_i, y_i)$ for all $i$. We remark that a good choice for $\pi^{\mathrm{ref}}$ is a cost-efficient model which is capable of producing diverse responses with positive reward, *e.g.*, a distilled version of a large reasoning model.

We train a classifier $f_\theta : \mathcal{S} \mapsto \Delta([K])$ via gradient descent on the following loss on data batch $\mathcal{B}$:

$$L(\theta; \mathcal{B}) = \frac{1}{|\mathcal{B}|} \sum_{(x_i, y_i, z_i, \kappa_i) \in \mathcal{B}} \frac{1}{|z_i|} \sum_{h \in \mathrm{range}(|z_i|)} \ell_{\mathrm{ce}}(f_\theta(x_i, y_i, z_i^{:h}), \kappa_i),$$

where $\ell_{\mathrm{ce}}(\hat{p}, \kappa) = -\ln(\hat{p}[\kappa])$ is the standard cross-entropy loss for classification and $z_i^{:h}$ denotes the first $h$ tokens of $z_i$. The rationale for the inner average is analogous to next-token prediction training of autoregressive models: since $z_i$ is generated autoregressively by $\pi^{\mathrm{ref}}$, any suffix $z_i^{h:}$ is also a roll-out from $\pi^{\mathrm{ref}}$ and hence can be viewed as another data-point. We found this to be an important training detail for performance, which is consistent with prior work who used a similar objective for training an outcome reward model [14, 24].

We can now view the classifier as a value model. Since $\kappa = 1$ corresponds to the event that $r = 1$, we have that $V_\theta(x) := f_\theta(x)[1]$ predicts the correctness probability of roll-outs from

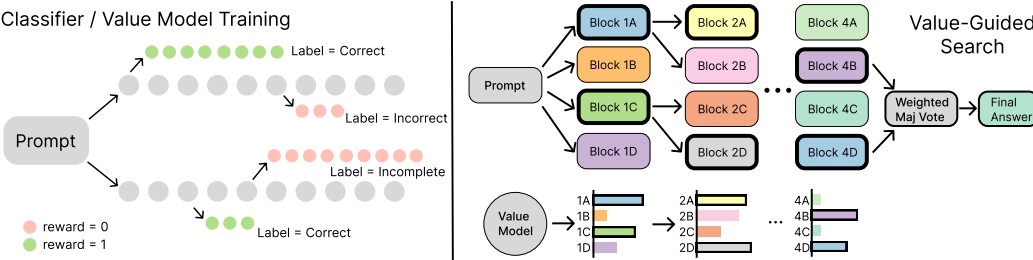

Figure 2: **Summary of Methods.** (Left) Diagrams how we collect multiple roll-ins (grey circles representing tokens) per problem, and branch off multiple roll-outs per roll-in at random points. The class label for each roll-out token is the outcome label at the very end. (Right) Shows the beam search process (beam width 2 and budget 4) guided by a value model.

| **Algorithm 1** Beam Search with Width $w$ | **Algorithm 2** Best-of-$N$ |
|---|---|
| 1: **Input:** prompt $x$. | 1: **Input:** prompt $x$, responses $\{y_i\}_{i\in[N]}$. |
| 2: Set num beams $B \leftarrow \frac{N}{w}$. | 2: **return** $y^{\text{bon}} = \arg\max_{y_i} V(x, y_i)$. |

**Algorithm 1** Beam Search with Width $w$

1: **Input:** prompt $x$.
2: Set num beams $B \leftarrow \frac{N}{w}$.
3: Initialize beams $y_1, \ldots, y_B \leftarrow x$.
4: **while** $\exists j$ s.t. $y_j$ is not finished **do**
5:    For each $j$ s.t. $y_j$ is not finished, sample $w$ *i.i.d.* blocks $\{b_{i,j}\}_{i\in[w]}$ from $\pi(\cdot \mid y_j)$.
6:    Update unfinished beams to be the best continuations with the highest $V(y_j, b_{i,j})$.
7: **end while**
8: **return** BoN or WMV on $\{y_1, \ldots, y_B\}$.

**Algorithm 2** Best-of-$N$

1: **Input:** prompt $x$, responses $\{y_i\}_{i\in[N]}$.
2: **return** $y^{\text{bon}} = \arg\max_{y_i} V(x, y_i)$.

**Algorithm 3** (Weighted) Majority Vote

1: **Input:** prompt $x$, responses $\{y_i\}_{i\in[N]}$, weights $\{w_i\}_{i\in[N]}$, equivalence relation $\sim$.
2: Partition $\{y_i\}_i$ into equiv. classes $\{p_k\}_k$.
3: **return** A response from the highest weight partition $\arg\max_{p_k} \sum_{y_i \in p_k} w_i$.

$\pi^{\text{ref}}$. Indeed, if $f^\star$ denotes the optimal classifier that minimizes the population-level loss, then $f^\star(x,y)[1] = \mathbb{E}_{z\sim\pi^{\text{ref}}(\cdot|x,y)}[R(x,y,z) \mid x,y]$ which is the expected reward of a completed response from rolling-out $\pi^{\text{ref}}$ starting from $x, y$. In sum, our value model is learned via predicting labels (one of which corresponds to reward 1), and the training objective is the standard cross-entropy loss.

## 2.2 Dataset Creation Process

We describe our process for creating `OpenR1-VM`, a novel dataset of 2.5 million reasoning responses from DeepSeek models, across 45k math problems from `OpenR1-Math` [2].

**Pre-Filtering.** We start from the `OpenR1-Math` dataset (default split) [2] which contains 94k math problems with solutions that were already filtered for quality. Upon manual inspection, we found that this dataset still contained unsolvable problems (*e.g.*, problems that require web browsing but our models cannot access the web) and ambiguous or unverifiable answers (*e.g.*, multiple \boxed{} expressions or unparsable answers). We filter out all such problematic problems, producing a cleaned subset of 50k problems with solutions verifiable by `sympy` or `math-verify` [23]. We call this pre-filtered dataset `OpenR1-Cleaned`.

**Response Generation.** Next, we collect roll-ins and roll-outs from DeepSeek models [19]. We fix the roll-out policy $\pi^{\text{ref}}$ as `DeepSeek-R1-Distill-Qwen-1.5B`. To ensure diversity in the roll-in distribution, we sample 14 independent roll-ins from four `DeepSeek-R1-Distill-Qwen` model sizes: 1.5B, 7B, 14B, and 32B by generating until the end of thinking token `<\think>`.[1] For each roll-in $\tilde{y}_i$, we then sample four random prefix locations where we generate complete roll-outs $\{z_i^j\}_{j\in[4]}$ from $\pi^{\text{ref}}$. Finally, to compute the class label (incomplete, incorrect, or correct), we parse the response for the final answer (enclosed in \boxed{}) and use `math-verify` to check for correctness against the ground truth answer. In total, this process (illustrated in Fig. 2 left) yields 56 labeled roll-in, roll-out pairs per problem, leading to 2.8 million datapoints.

**Post-Filtering.** We filter out problems where all 56 roll-outs for that problem were incomplete or incorrect (*i.e.*, has reward 0). This post-filtering removes any ambiguous or unanswerable problems that we missed during pre-filtering, and also removes problems that are too difficult for $\pi^{\text{ref}}$ and do not provide a useful learning signal. This step filters roughly 10% of problems, yielding a final dataset of 2.5 million datapoints.

Notably, our approach does not require a fine-grained notion of step and our data collection is cheaper than existing PRM techniques. Specifically, Lightman et al. [24] used per-step annotations by human experts, Zhang et al. [51] used per-step annotations via LLM-as-a-Judge, and Wang et al. [45] used multiple Monte Carlo roll-outs at every step. Since the number of newlines in reasoning CoT traces can grow very quickly, per-step labels are quite expensive to collect for reasoning models. In contrast, our approach only requires a handful of roll-ins (from any policy) and roll-outs (from $\pi^{\text{ref}}$) per problem, and this number can be flexibly tuned up or down to trade-off data coverage and data collection cost. Please refer to Appendix D for further details on each step. We also release our filtering code and datasets to support future research.

---

[1] `DeepSeek-R1` and its distilled variants output CoT reasoning between tokens `<think>` and `<\think>` followed by a final solution, which is usually a summarization of the CoT reasoning.

## 2.3 Algorithms for Test-Time Compute and Search

Equipped with a value model $V : \mathcal{S} \mapsto \mathbb{R}$, we can now apply it to scale test-time compute of a generator model $\pi$. For search-based approaches, we focus on block-wise search where a "block" refers to a sequence of tokens (*e.g.*, blocks of 4096 tokens worked best in our experiments). We let $N$ denote the inference budget, which is the number of generations we can sample (*e.g.*, generating four responses and taking a majority vote is $N = 4$).

**BFS.** Breadth-first-search (BFS) [48, 29] is a natural search method that approximates the optimal KL-regularized policy given a good value model [53]. Given a prompt $x$, BFS samples $N$ parallel blocks $b_i$ from $\pi$ and selects the block with the highest value $b^\star = \arg\max_{b_i} V(x, b_i)$, which gets added to the prompt, *i.e.*, $x \leftarrow x, b^\star$. The process repeats until the response finishes. Note the number of tokens generated from $\pi$ is roughly equivalent to $N$ independent generations from $\pi$.

**Beam Search.** One weakness of BFS is that parallel blocks are correlated because they share the same prefix, which limits diversity. Beam search with width $w$ (Algorithm 1) is a generalization that keeps $B = N/w$ (assume to be integer) partial responses and branches $w$ parallel blocks from each one [27, 5, 41, 6, 37]. Given a prompt $x$, beam search first generates $N$ parallel blocks. However, unlike BFS, beam search keeps the top $B$ beams with the highest scores, and then samples $w$ parallel blocks per beam at the next step. Since $B \times w = N$ blocks are sampled at each step, the compute budget is also $N$. We illustrate beam search with $N = 4$ and $w = 2$ in Fig. 2 (right).

**DVTS.** Diverse verifier tree search (DVTS) is a meta-algorithm that further increases diversity by running parallel searches each with smaller budgets [6]. Specifically, DVTS-$M$ runs $M$ parallel beam searches each with budget $N/M$ (assume to be integer), and aggregates responses into a final answer.

We remark a crucial detail of beam search and DVTS is how the final set of beams/responses are aggregated. Prior works [6, 37, 35] select the response with the highest score, which is analogous to a final best-of-$N$ (BoN; Algorithm 2). Instead, we found that taking a weighted majority vote (WMV; Algorithm 3) led to much better performance, which is demonstrated by Fig. 3 (left).

**Computational Efficiency of Block-wise Search.** Since value scores are only used at the end of each block or the end of the whole response, the FLOPs required for block-wise value model guidance is a tiny fraction ($\ll 1\%$) of the generation cost from $\pi$. We compute FLOP estimates in Appendix H to concretely show this.

# 3 Experiments

We extensively evaluate value-guided search (VGS) with our 1.5B value model `DeepSeek-VM-1.5B`, focusing on guiding the CoT reasoning of DeepSeek models [19]. The best VGS setup for our value model is beam search with final WMV aggregation, beam width 2, block size 4096 and with DVTS (for larger inference budgets). We show this setup outperforms other test-time compute methods (*e.g.*, MV, WMV, BoN) and other scoring models (*e.g.*, existing 7B PRMs and a 1.5B Bradley-Terry reward model trained on our dataset). *We remark our search results use a fixed beam width and block size for all problems*; this is more practical than prior works on "compute-optimal scaling" which vary search parameters for each problem and require estimating each problem's difficulty [6, 37, 26]. Please see Appendices E and F for additional details on value model training and inference.

**Benchmarks.** We evaluate on the American Invitational Mathematics Exam (AIME) and the February Harvard-MIT Mathematics Tournament (HMMT).[2] Both AIME and HMMT are prestigious high school math competitions in the US that have also been used to evaluate frontier LLMs [32, 19, 1]. We use AIME I & II and the individual part of HMMT, yielding 30 problems per competition. We selected the block size and beam width based on the performance AIME-24 as the validation set (ablations in Section 4.1). Then, we evaluate on AIME-25 and HMMT-25 since they happened after the release of DeepSeek and OpenR1. This controls for data contamination since the test problems were released after the underlying reference models and datasets. We also ensure that there are no problems with $> 50\%$ 8-gram overlap between training and test sets as a further sanity check for contamination. Unless otherwise stated, we report the overall accuracy averaged across AIME-25 and HMMT-25. In Appendix C, we report individual per-benchmark results for the 2024 and 2025 editions of both math competitions.

---

[2] https://maa.org/maa-invitational-competitions and https://www.hmmt.org

| Test-time scaling `DeepSeek-1.5B` ($N = 256$) | AIME-25 | HMMT-25 | AVG |
|---|---|---|---|
| VGS w/ `DeepSeek-VM-1.5B` (ours) | $46.7 \pm 0.7$ | $32.8 \pm 0.8$ | $\mathbf{39.8 \pm 0.5}$ |
| WMV w/ `DeepSeek-VM-1.5B` (ours) | $45.1 \pm 2.2$ | $28.9 \pm 2.6$ | $\underline{37.0 \pm 1.7}$ |
| VGS w/ `DeepSeek-BT-1.5B` (ours) | $40.6 \pm 0.8$ | $27.5 \pm 0.0$ | $34.1 \pm 0.4$ |
| WMV w/ `DeepSeek-BT-1.5B` (ours) | $40.5 \pm 2.9$ | $24.6 \pm 4.7$ | $32.6 \pm 1.6$ |
| VGS w/ `Qwen2.5-Math-PRM-7B` | $38.9 \pm 1.4$ | $24.2 \pm 0.2$ | $31.6 \pm 0.7$ |
| WMV w/ `Qwen2.5-Math-PRM-7B` | $39.1 \pm 2.1$ | $24.0 \pm 3.2$ | $31.6 \pm 1.9$ |
| VGS w/ `MathShepherd-PRM-7B` | $41.9 \pm 1.4$ | $23.9 \pm 1.4$ | $32.9 \pm 1.0$ |
| WMV w/ `MathShepherd-PRM-7B` | $40.0 \pm 2.5$ | $25.6 \pm 3.1$ | $32.8 \pm 2.0$ |
| MV@256 | $38.9 \pm 1.9$ | $24.3 \pm 2.9$ | $31.6 \pm 1.7$ |
| Test-time scaling `DeepSeek-7B` ($N = 128$) | | | |
| VGS w/ `DeepSeek-VM-1.5B` | $59.4 \pm 0.8$ | $41.1 \pm 1.6$ | $\mathbf{50.3 \pm 0.9}$ |
| MV | $56.5 \pm 1.6$ | $33.8 \pm 2.5$ | $\underline{45.2 \pm 1.5}$ |
| Pass@N baselines for various models | | | |
| `DeepSeek-1.5B` Pass@1 | $22.4 \pm 4.1$ | $13.0 \pm 3.9$ | $17.7 \pm 2.8$ |
| `DeepSeek-32B` Pass@1 | $60.4 \pm 6.0$ | $42.1 \pm 5.2$ | $\underline{51.3 \pm 4.0}$ |
| `Deepseek-R1` (671B) Pass@1 | $70.0 \pm 0.9$ | $46.7 \pm 2.4$ | $\mathbf{58.4 \pm 1.3}$ |

Table 1: (Top) Weighted majority vote (WMV) and `VGS` results for `DeepSeek-1.5B` with an inference budget of $N = 256$, using various scoring models. (Middle) Compares MV and `VGS` for larger DeepSeek models guided with our `DeepSeek-VM-1.5B`. (Bottom) Lists performance of DeepSeek models and strong close-sourced reasoning models. For `VGS`, $\pm$ indicates standard deviation across 3 seeds; for MV, WMV, Pass@N, $\pm$ denotes bootstrap with 100 repetitions. We **bold** the highest avg. accuracy and underline second highest. Appendix C.1 contains more baselines.

**Baseline Models.** We evaluate two state-of-the-art 7B PRMs with distinct training styles: `Math-Shepherd-Mistral-7B-PRM` [45] and `Qwen2.5-Math-PRM-7B` [51]. Math-Shepherd uses Monte-Carlo roll-outs from each step to estimate per-step value while the Qwen2.5 PRM uses LLM-Judge annotation for each step, similar to the per-step human annotation of PRM800K [24]. As a step-level value model, `Math-Shepherd-PRM-7B` is more related to our token-level value model. Finally, we also evaluate a 1.5B Bradley-Terry (BT) [8] model, called `DeepSeek-BT-1.5B`, which we trained using our dataset (see Appendix G for training details).

## 3.1 Main Results (Table 1)

In the top section of Table 1, we fix the generator to `DeepSeek-1.5B`[3] and test-time budget to $N = 256$, and compare `VGS` to weighted majority voting (WMV), using our value model, the BT model and baseline PRMs. We see that `VGS` and WMV with `DeepSeek-VM-1.5B` achieve the two highest scores, outperforming the BT reward model and prior PRMs. This shows that our value model is not only a strong outcome reward model (ORM) but also an effective value model for guiding search. Intriguingly, while `DeepSeek-BT-1.5B` was only trained as an ORM, we find that `VGS` also improves performance relative to WMV, suggesting that BT models may also provide meaningful block-wise feedback to guide search. We also observe that accuracies for the 7B baseline PRMs (MathSheperd and Qwen2.5-Math) are only slightly higher than `MV@256` and do not improve with search, which suggests that these PRMs are likely out-of-distribution (OOD) for the long CoTs generated by `DeepSeek-1.5B`.

In the middle section of Table 1, we guide the stronger 7B DeepSeek model and compare `VGS` to MV, a standard TTC method that does not use an external scoring model. We see that `VGS` again achieves higher accuracy than MV for 7B, which suggests that `DeepSeek-VM-1.5B` is also useful for guiding CoT of larger DeepSeek models. In Table 2, we also include results for guiding 14B DeepSeek, where we observe that the gap between `VGS` and MV becomes much smaller. This suggests that `DeepSeek-14B` CoTs may be becoming OOD for our value model trained on `DeepSeek-1.5B` CoTs. To guide more capable models, new value models should be trained on rollouts from similarly capable models; we however do not foresee this being a practical concern given the scalability of our training process (described in Section 2 and summarized in Appendix B).

---

[3]Throughout the paper, we use `DeepSeek-XB` as shorthand for `DeepSeek-R1-Distill-Qwen-XB`.

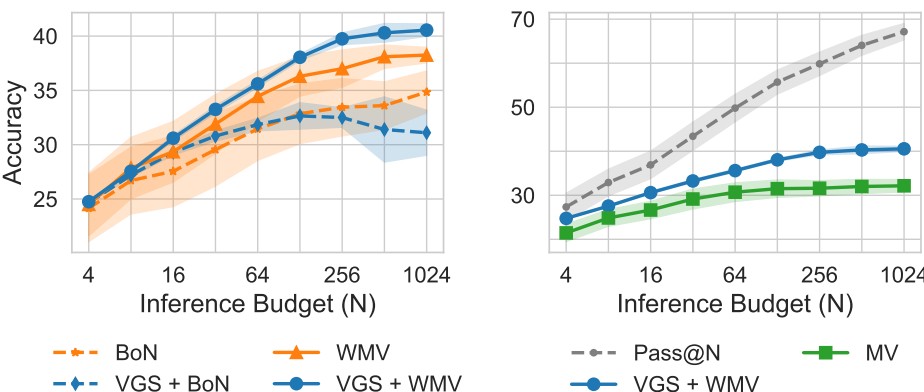

Figure 3: **Test-Time Compute with DeepSeek-VM-1.5B.** (Left) Compares best-of-$N$ (BoN), weighted majority voting (WMV) and VGS with either BoN or WMV for the final aggregation. (Right) Compares VGS to majority voting (MV), a standard baseline that does not require a scorer.

## 3.2 Test-Time Compute Scaling for Search

This section presents three experiments designed to analyze the TTC scaling properties of VGS. Our investigation addresses three key research questions:

1. Does VGS, with its block-wise guidance, demonstrate superior performance compared to response-level aggregation methods such as BoN or WMV?

2. How does the TTC scaling behavior of VGS compare to the standard score-free baseline MV?

3. How does the TTC scaling behavior of DeepSeek-VM-1.5B compare to baseline models?

**Response-Level Selection vs Search-Based Block-Level Selection.** While BoN and WMV represent standard approaches for selecting responses using an ORM, block-wise VGS guides response generation through sequential block-by-block selection. As Fig. 3 (left) illustrates, WMV consistently outperforms BoN across all inference budget scales, which demonstrates the benefits of combining MV with value scores. Furthermore, VGS (with WMV as a final aggregation step) yields additional improvements beyond WMV alone. This confirms the benefits of search and aligns with conclusions from previous studies [6, 37, 26]. Interestingly, we do not observe the same benefits of search if BoN is used as a final aggregation step, suggesting that WMV is a critical component to VGS.

**Response Length for VGS.** In addition to consistent performance gains, VGS also produces noticeably shorter responses compared to the base DeepSeek-1.5B model. In Figure 15 (Appendix C.7), we present histograms of response lengths across all benchmarks. The results show that VGS consistently generates more concise outputs, whereas the base model often reaches the generation cap, with up to

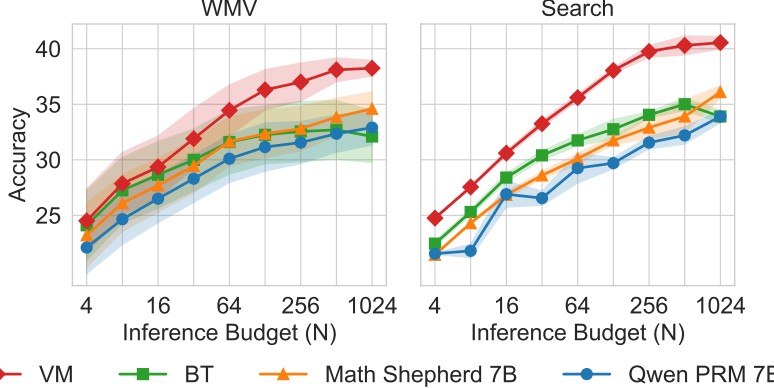

Figure 4: **TTC Scaling of Various Scorers.** Comparison of our 1.5B value model (VM), our 1.5B Bradley-Terry reward model (BT), and two 7B state-of-the-art PRMs for two TTC scaling methods: (Left) WMV or (Right) VGS (with WMV as a final aggregation step).

50% of its responses being unfinished. On average, VGS responses are 11,219 tokens long, compared to 12,793 for DeepSeek-1.5B, representing a reduction of over 12% in token and thus FLOPs usage.

**VGS vs Majority Voting.** As Fig. 3 (right) demonstrates, VGS consistently achieves higher accuracy than MV, attaining equivalent performance while requiring substantially lower inference budgets (also shown in Fig. 1 right). Fully closing the gap with the oracle Pass@N curve may require a larger value model trained on more extensive datasets.

**DeepSeek-VM-1.5B vs Baseline Scoring Models.** Fig. 4 benchmarks DeepSeek-VM-1.5B against existing PRMs and our BT model. We observe that DeepSeek-VM-1.5B consistently delivers superior performance when employed both as an ORM for WMV (left) and as a guidance mechanism for block-wise search (right). Note that we find our BT model to be surprisingly effective as a search guidance model which suggests the importance of our token-level dataset playing an important role in successful downstream search.

### 3.3 Scaling up the Generator Model Sizes

In Fig. 5, we scale up our experiments to guide larger 7B and 14B DeepSeek models. Here, we run VGS with the same search parameters using the same value model DeepSeek-VM-1.5B. Although the 7B and 14B DeepSeek rollins are OOD for our value model (trained on 1.5B traces), we observe that VGS continues to scale without plateauing as test-time compute increases. This provides some evidence that a value model trained with a weaker verifier policy can generalize effectively and guide the CoTs of stronger models, which is useful since it is much cheaper to collect training data from smaller $\pi^{\text{ref}}$ models. This form of "weak-to-strong" generalization [10] appears to be a promising direction for future research.

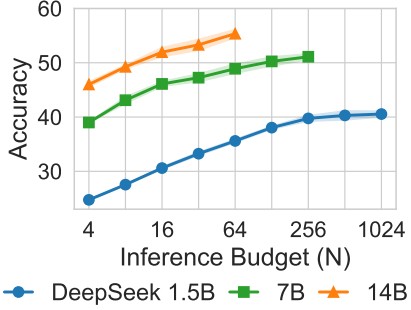

Figure 5: **VGS + WMV Performance when Guiding Larger Models.** With the same DeepSeek-VM-1.5B providing guidance, search continues to improve with more test-time compute.

### 4 Ablation Studies

To investigate the role of key hyperparameters in value-guided search, we perform ablation analyses of block size and beam width on AIME-24 across varying inference budgets. We also ablate the amount of DVTS parallelism. These tests suggest that there is a consistent choice of search hyperparameters that work well across inference budgets.

### 4.1 Different Search Parameters and Methods

**Block Size.** We perform beam search with width 2 using search block sizes from 16 to 16384. Fig. 6 shows AIME-24 accuracies across three inference budgets $N$, revealing that the optimal choice of 4096 stays consistent across different $N$. We see a decline in performance when searching with more fine-grained blocks.

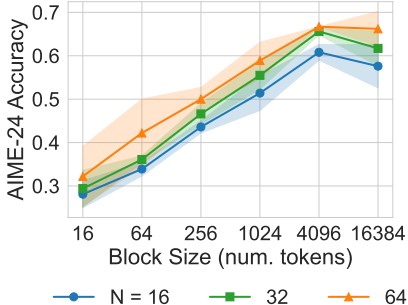

Figure 6: **Ablation: Search Block Size.** AIME-24 accuracies for beam search (width 2) with varying block sizes from 16 to 16384. We found 4096 to be optimal across test-time budgets and benchmarks.

**Beam Width.** We perform beam search with block size 4096 using varying beam widths, with breadth-first-search (BFS) being a special case where beam width is equal to $N$. Fig. 7 (left) shows AIME-24 accuracies across five inference budgets, demonstrating that beam width 2 is consistently optimal across different $N$. We note our optimal beam width is different from prior works' which found 4 to work best [6, 37, 26].

**DVTS Parallelism.** Fig. 7 (right) shows the role of ablating DVTS from VGS. For each inference budget, we report average accuracies without DVTS and with the best DVTS parallelism $M$. We observe that DVTS becomes more effective at higher budgets and scales better than a single search

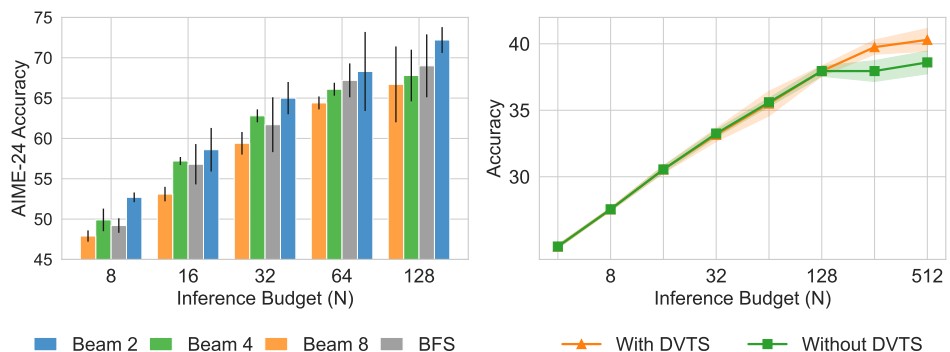

Figure 7: **Ablations: Beam-Width and DVTS.** (Left) AIME-24 accuracies for beam search with various widths (Section 2.3) across inference budgets $N$. BFS is equivalent to setting width as $N$. We find that the optimal beam width is robust across multiple TTC budgets. (Right) Averaged accuracy for beam 2 with and without DVTS. For DVTS, we report the best result with parallelism $M > 1$ per inference budget $N$, which we find scales better at higher budgets.

tree, which is consistent with findings from prior works [6]. However, we find that DVTS is never worse than a single search tree even at smaller inference budgets, which is the opposite conclusion reached by prior works [6]. This discrepancy may be explained by the fact that we use WMV to combine the DVTS responses, which seems to be a more robust way to perform DVTS than BoN (used in prior works) given our findings from Fig. 3.

## 4.2 Random vs. Value-Guided Search

Finally, we directly ablate the role of our value model's guidance during the search process. We perform VGS (w/ same width, block size and DVTS) but randomly select blocks instead of selecting blocks with the highest value. We still aggregate the final beams via WMV with our value model, so the only change is how intermediate blocks are chosen. We call this process "random search". Thus, if our value model is helpful for search, we should expect VGS to outperform random search. Indeed, Fig. 8 validates this hypothesis. We also evaluate a hybrid approach where half of DVTS's parallel trees use random search and the other half use VGS. We find that this hybrid approach lands roughly between pure VGS and pure random search, again validating that block-selection from our value model improves over random selection.

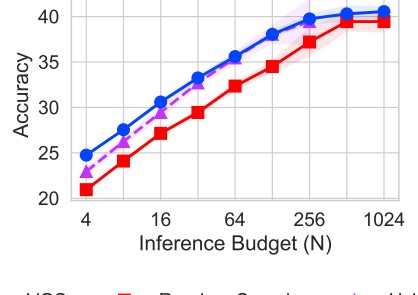

Figure 8: **Ablation: Random Search.** Random search is the same search process as VGS except intermediate blocks are randomly selected instead of using our value model. Hybrid is a mixture where we flip a fair coin at the start of a search tree that decides whether to use random search or VGS. We see that selecting blocks with highest value improves accuracy across inference budgets.

## 5 Conclusion

In this paper, we introduced block-wise *Value-Guided Search* (VGS), a simple yet effective strategy for steering long-context CoT reasoning models. We proposed a scalable token-level value model training pipeline that does not require a pre-defined notion of "step" or expensive per-step annotations. We collect a large dataset of reasoning CoTs (OpenR1-VM) and train a lean 1.5B value model (DeepSeek-VM-1.5B), which we show can effectively guide the CoTs of DeepSeek models up to 14B in size. With extensive experiments, we demonstrate that VGS with DeepSeek-VM-1.5B enjoys better test-time compute scaling than standard methods (*e.g.*, majority voting, best-of-$N$) and other scoring models (*e.g.*, existing PRMs and a BT model), achieving a higher performance ceiling while reducing the FLOPs needed to extract the same performance as baseline methods (Fig. 1). Our results point to VGS as a promising approach to scale TTC of emerging reasoning models.

**Discussion of Limitations.** Our value model is trained exclusively on completions / roll-outs from a lean reasoning model $\pi^{\mathrm{ref}}$ (*e.g.*, DeepSeek-R1-Distill-Qwen-1.5B). As frontier LLMs continue

to advance, the distribution of their generated responses may increasingly diverge from our training distribution, potentially degrading scoring and search performance. To maintain optimal performance, new value models will need to be retrained on rollouts from updated generator policies. However, we do not foresee this as a major practical concern given the simplicity and scalability of our pipeline. To facilitate retraining and adaptation to similar verifiable domains, we open-source our codebase and provide a step-by-step recipe in Appendix B for data collection, training and search inference.

## Acknowledgment

KW is supported by a Google PhD Fellowship. JPZ is supported by a grant from the Natural Sciences and Engineering Research Council of Canada (NSERC) (567916). ZG is supported by LinkedIn-Cornell Grant. Wen Sun is supported by NSF IIS-2154711, NSF CAREER 2339395 and DARPA LANCER: LeArning Network CybERagents. This research is also supported by grants from the National Science Foundation NSF (IIS-1846210, IIS-2107161, and IIS-1724282, HDR-2118310), the Cornell Center for Materials Research with funding from the NSF MRSEC program (DMR-1719875), DARPA, arXiv, LinkedIn, Google, and the New York Presbyterian Hospital.

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

# Appendices

**Table of Contents**

Note: In the appendix, we also provide additional empirical results in Appendix C. Two new results are worth highlighting here. First, in Appendix C.6, we provide test-time scaling results for guiding `DeepSeek-1.5B` further trained with PPO on our math dataset. We find that `VGS` improves test-time scaling compared to MV and WMV, which shows that our method nicely complements policy-based RL training. Moreover, in Appendix C.7, we include three qualitative examples of contrastive blocks that were selected or rejected by our value model during beam search process. We see that our value model prefers blocks with more straightforward logical deductions, yielding more efficient and effective CoT for reasoning.

## A   Related Works

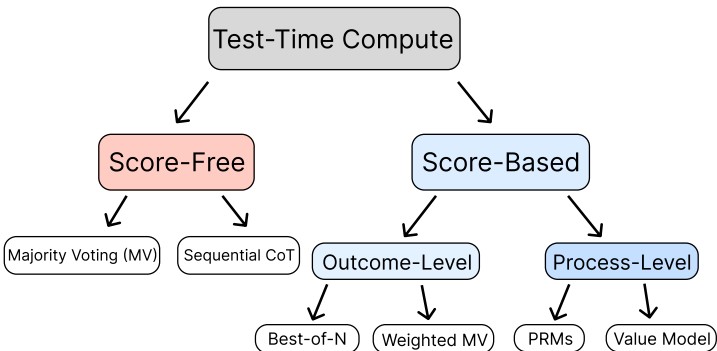

Figure 9: **Taxonomy of TTC Methods.** Score-free TTC methods do not require an external scoring model, *e.g.*, by taking a majority vote. Score-based TTC methods require an external scoring model. The coarsest scoring model is an outcome reward model (ORM), which scores a whole response and can be used for best-of-$N$ or weighted MV. A more fine-grained scoring model are process-level scorers, which includes process reward models (PRMs) and value models; these more fine-grained scoring models can be used for search.

Test-time compute (TTC) broadly refers to algorithms that improve problem-solving performance when given more compute (*i.e.*, FLOPs) at test-time. Fig. 9 summarizes the taxonomy of TTC methods. The simplest TTC methods are score-free in the sense that they do not require access to an external scoring model. A notable example is majority voting (MV), which selects the most frequent answer among $N$ responses, breaking ties randomly [14, 47, 9]. Also known as self-consistency, MV can be applied to tasks where the output space is equipped with an equivalence relation, *e.g.*, mathematical formulae that can be symbolically checked for equality. Other score-free TTC methods include sequentially revising the response via CoT prompting [39, 30] and hybrid methods [42].

There are also score-based TTC methods that employ an external scorer. The coarsest type of a scorer is an *outcome reward model* (ORM), which takes the full prompt and response as input and produces a scalar that measures the quality / correctness of the response. Popular examples of ORMs include Bradley-Terry reward models [8] or LLM-as-a-Judge [4]. ORMs can be used for

best-of-$N$ (BoN), which selects the response with the highest score [14, 4]. ORMs can also be used for weighted majority voting (WMV), which generalizes MV where the strength of a response's vote is proportional to its ORM score. Weighted MV (WMV) typically provides an improvement over vanilla (unweighted) MV [6, 26], which is also what we observe in our experiments (Fig. 3).

Outcome-level TTC methods (*e.g.*, BoN, WMV) may be further refined with *process-level* scorers that guide the generation process in a fine-grained manner. We remark that our value model can act as both an outcome-level and a process-level scorer. When queried with a partial response, the value model predicts the expected quality of *future* completions under $\pi^{\text{ref}}$. When queried at the end of a full response, the value model predicts the quality of the final response. Indeed, our best performing setup for value-guided search (VGS) uses intermediate values to guide block-wise beam search and uses final values via WMV to aggregate the final beams, which employs both the process-level and outcome-level scoring capabilities of the value model. Finally, to the best of our knowledge, the combination of search with WMV is novel to this work, and we found this to be a crucial ingredient to effectively scale TTC of DeepSeek models.

Prior works on process-level TTC largely focused on *step-wise* search with *process reward models* (PRMs), which measures the correctness of a fine-grained step [24]. They showed that step-wise search can provide better performance than outcome-level TTC methods [29, 48, 25, 37, 49, 35, 13]. However, training step-wise PRMs requires a pre-defined notion of step, which is challenging to explicitly define for general reasoning [19]; *e.g.*, prior works used newlines \n to separate steps, but DeepSeek's CoTs often contain newlines in-between coherent thoughts. Moreover, prior PRM training techniques require per-step annotations, via humans [24], LLM-Judges [51], or per-step MC rollouts [45, 28], which are expensive to collect for long reasoning traces; *e.g.*, a single response from DeepSeek models typically contains hundreds of newlines.

These limitations make it difficult to scale PRMs to long-context reasoning models [19], and all these prior works could only evaluate on short-context models with easier benchmarks such as GSM8k [14] and MATH [24]. In contrast, our paper focuses on scaling process-level guidance to long-context reasoning models, and we propose a block-level search method that mitigates the above limitations. We train a token-level value model by collecting rollouts from random solution prefixes, which requires neither a pre-defined notion of step nor per-step annotations. We use our value model to guide a block-wise search process, where the block size is a hyperparameter and we find there exists a consistent choice that works well across inference budgets (Fig. 6). Crucially, we are able to scale our value-guided search (VGS) to long-context DeepSeek models and demonstrate impressive performance and efficiency gains on challenging math competitions (Fig. 1).

Closely related to our work is Setlur et al. [35], who propose to train a token-level process advantage verifier (PAV), which is the sum of $\pi^{\text{ref}}$'s $Q$-function and an (off-policy) expert's advantage function, to guide step-wise search. This method is similar to ours since the training process also occurs at a token-level and is agnostic to the definition of step. However, a limitation of the PAV is that if the expert disagrees with the underlying policy, then maximizing the PAV can lead to suboptimal behavior [12]. Our approach of directly using the value model does not have this issue. Moreover, Setlur et al. [35] proposed to use PAVs to guide *step-wise* search, which still requires a definition of step at inference time; in contrast, we propose to use block-wise search which does not require a definition of step at inference. At a technical level, Setlur et al. [35] trained the PAV by minimizing the mean-squared error (MSE) loss; in contrast, we propose to use the cross-entropy loss, which has been shown to work better for downstream decision making [17, 44, 53].

We remark that some prior works proposed to use token-level value models to reweight the *next-token* distribution of the generator [29, 50, 20, 53]. However, these methods require one classifier call per token, which is more expensive than block-wise search. Moreover, token-level guidance might also be less effective because the imperfect value model may introduce cascading errors if queried at every token. We highlight that Mudgal et al. [29] also experimented with block-wise BFS and found this to be more effective at scaling test-time compute than reweighting the next-token distribution (*i.e.*, token-wise guidance). One drawback of block-wise BFS is that the blocks may all become correlated due to sharing the same prefix. Thus, we build upon Mudgal et al. [29] by proposing to use beam search, which we show yields better test-time scaling for reasoning models (Fig. 6).

Recently, Fu et al. [18] showed that confidence as measured by the average next-token entropy along the thinking trace can also be used to guide the CoT generation process to improve TTC performance. An advantage of Fu et al. [18] is that it does not require training a value model and can be applied

directly given a generator policy. However, it does not leverage any reward signals and may fall short in planning long-term strategies [53]. It is a promising future direction to explore how to combine uncertainty-based methods with value-based methods.

## B  Summary of VGS Pipeline

We provide a step-by-step recipe for running VGS on any verifiable domain of interest. This recipe is applicable to any task with a reward label for responses (*i.e.*, outcome-level feedback). If the task has continuous rewards, a standard trick from distributional RL is to discretize the reward distribution as a histogram, and then the value model is simply the expected reward under the learned distribution [7, 21, 17, 43, 44].

1. Start with a verifiable domain, where responses are identified with a label and a reward.

2. Identify a good dataset of prompts.

3. Identify a set of roll-in policies and a single roll-out policy. The roll-in policies should provide a diverse distribution of solutions, and the roll-out policy should be strong enough to complete responses given a partial roll-in.

4. For each prompt, sample $n$ roll-in responses from the set of roll-in policies.

5. For each roll-in response, sample $m$ random indices $\{i_j\}_{j\in[m]}$, and collect a roll-out per index. Thus, there are $nm$ roll-in, roll-out pairs per prompt.

6. Post-filter by removing prompts where all roll-out responses fail to complete or solve the prompt.

7. At this point, we have created our dataset of roll-in, roll-out pairs. We are now ready to train our value model.

8. Train a classifier / value model by following (Section 2.1). Sweep hyperparameters such as learning rate.

9. Choose a generator policy to be guided by the value model. The most in-distribution choice is to use the roll-out policy $\pi^{\text{ref}}$.

10. Perform model selection by running outcome-level TTC (*e.g.*, WMV) on some validation benchmark.

11. Sweep search parameters (*e.g.*, block size, beam width, DVTS parallelism) on the validation benchmark.

12. Run the final model on the test benchmark with the best search parameters.

The sampling distribution for the cut-off index (Step 5) is also worth tuning. For example, values at earlier or middle indices may be harder to predict than final indices, so it is worth sampling more cut-off indices from these earlier regions.

# C Additional Experiment Results

## C.1 Full Main Results Table

We reproduce Table 1 with additional results and baselines. The results with closed source models were obtained via API calls in May 2025. We see that with a budget of 256, our 1.5B value model can guide `DeepSeek-1.5B` (total parameter count is 3B) to reach parity with the pass@1 of `o3-mini-medium`.

| Test-time scaling `DeepSeek-1.5B` ($N = 256$) | AIME-24 | AIME-25 | HMMT-24 | HMMT-25 | AVG |
|---|---|---|---|---|---|
| VGS w/ `DeepSeek-VM-1.5B` (ours) | $72.0 \pm 0.4$ | $46.7 \pm 0.7$ | $31.4 \pm 2.0$ | $32.8 \pm 0.8$ | $\mathbf{45.7 \pm 1.0}$ |
| WMV w/ `DeepSeek-VM-1.5B` (ours) | $69.6 \pm 3.9$ | $45.1 \pm 2.2$ | $29.1 \pm 2.6$ | $28.9 \pm 2.6$ | $\underline{43.2 \pm 1.4}$ |
| VGS w/ `DeepSeek-BT-1.5B` (ours) | $73.1 \pm 1.4$ | $40.6 \pm 0.8$ | $28.1 \pm 1.9$ | $27.5 \pm 0.0$ | $42.3 \pm 0.5$ |
| WMV w/ `DeepSeek-BT-1.5B` (ours) | $72.0 \pm 3.3$ | $40.5 \pm 2.9$ | $25.3 \pm 2.3$ | $24.6 \pm 4.7$ | $40.6 \pm 1.6$ |
| VGS w/ `Qwen2.5-Math-PRM-7B` | $71.1 \pm 1.0$ | $38.9 \pm 1.4$ | $26.7 \pm 1.2$ | $24.2 \pm 0.2$ | $40.2 \pm 0.5$ |
| WMV w/ `Qwen2.5-Math-PRM-7B` | $70.6 \pm 3.1$ | $39.1 \pm 2.1$ | $25.4 \pm 2.4$ | $24.0 \pm 3.2$ | $39.8 \pm 1.4$ |
| VGS w/ `MathShepherd-PRM-7B` | $70.6 \pm 3.1$ | $41.9 \pm 1.4$ | $30.0 \pm 1.4$ | $23.9 \pm 1.4$ | $41.6 \pm 0.9$ |
| WMV w/ `MathShepherd-PRM-7B` | $71.2 \pm 3.2$ | $40.0 \pm 2.5$ | $27.9 \pm 2.3$ | $25.6 \pm 3.1$ | $41.2 \pm 1.4$ |
| MV@256 | $71.0 \pm 3.5$ | $38.9 \pm 1.9$ | $24.4 \pm 1.7$ | $24.3 \pm 2.9$ | $39.7 \pm 1.2$ |
| Test-time scaling larger models with our `DeepSeek-VM-1.5B` | | | | | |
| VGS w/ `DeepSeek-7B` ($N = 128$) | $82.2 \pm 0.8$ | $59.4 \pm 0.8$ | $42.8 \pm 2.8$ | $41.1 \pm 1.6$ | $\mathbf{56.4 \pm 0.8}$ |
| MV w/ `DeepSeek-7B` ($N = 128$) | $77.1 \pm 1.1$ | $56.5 \pm 1.6$ | $34.7 \pm 1.6$ | $33.8 \pm 2.5$ | $\underline{50.5 \pm 0.9}$ |
| VGS w/ `DeepSeek-14B` ($N = 64$) | $86.7 \pm 2.7$ | $59.6 \pm 0.6$ | $46.7 \pm 2.7$ | $51.1 \pm 1.6$ | $\mathbf{61.0 \pm 0.9}$ |
| MV w/ `DeepSeek-14B` ($N = 64$) | $80.6 \pm 1.2$ | $67.0 \pm 2.0$ | $40.6 \pm 1.8$ | $50.1 \pm 2.0$ | $\underline{59.6 \pm 0.9}$ |
| Pass@N baselines for various models | | | | | |
| `DeepSeek-1.5B` Pass@1 | $28.2 \pm 6.1$ | $22.4 \pm 4.1$ | $13.9 \pm 4.2$ | $13.0 \pm 3.9$ | $19.4 \pm 1.1$ |
| `DeepSeek-1.5B` Pass@256 | $81.9 \pm 1.7$ | $62.6 \pm 3.6$ | $54.2 \pm 4.9$ | $57.1 \pm 3.8$ | $63.9 \pm 0.9$ |
| `DeepSeek-7B` Pass@1 | $54.8 \pm 6.0$ | $40.9 \pm 6.1$ | $31.5 \pm 4.4$ | $25.5 \pm 4.6$ | $38.2 \pm 1.3$ |
| `DeepSeek-14B` Pass@1 | $72.4 \pm 5.4$ | $53.9 \pm 5.5$ | $36.4 \pm 4.8$ | $36.5 \pm 5.5$ | $49.8 \pm 1.3$ |
| `DeepSeek-32B` Pass@1 | $77.2 \pm 4.9$ | $60.4 \pm 6.0$ | $38.0 \pm 4.6$ | $42.1 \pm 5.2$ | $\underline{54.4 \pm 1.3}$ |
| `Deepseek-R1` (671B) Pass@1 | $85.0 \pm 2.1$ | $70.0 \pm 0.9$ | $41.7 \pm 3.5$ | $46.7 \pm 2.4$ | $\mathbf{60.8 \pm 0.5}$ |
| `o1-mini-medium` Pass@1 | $63.3 \pm 6.6$ | $52.3 \pm 6.8$ | $33.1 \pm 5.1$ | $34.0 \pm 5.9$ | $45.7 \pm 1.5$ |
| `o1-mini-medium` Pass@8 | $83.7 \pm 2.7$ | $81.8 \pm 3.7$ | $58.0 \pm 4.0$ | $52.8 \pm 3.4$ | $69.1 \pm 1.7$ |
| `o3-mini-medium` Pass@1 | $49.2 \pm 6.8$ | $45.8 \pm 6.6$ | $32.4 \pm 5.4$ | $36.6 \pm 6.0$ | $41.0 \pm 1.5$ |
| `o3-mini-medium` Pass@8 | $83.0 \pm 4.6$ | $77.4 \pm 3.9$ | $55.9 \pm 4.3$ | $64.9 \pm 4.4$ | $\underline{70.3 \pm 2.1}$ |
| `o4-mini-medium` Pass@1 | $85.4 \pm 4.3$ | $82.3 \pm 4.5$ | $50.4 \pm 5.0$ | $61.1 \pm 6.4$ | $69.8 \pm 2.5$ |
| `o4-mini-medium` Pass@8 | $95.4 \pm 2.6$ | $93.3 \pm 0.4$ | $69.7 \pm 3.2$ | $84.5 \pm 2.5$ | $\mathbf{85.7 \pm 1.1}$ |

Table 2: (Top) Weighted majority vote (WMV) and `VGS` accuracies for `DeepSeek-1.5B` with an inference budget of $N = 256$, with various scoring models. (Middle) Compares MV and `VGS` for larger DeepSeek models guided with our `DeepSeek-VM-1.5B`. (Bottom) Lists performance of DeepSeek models and strong close-sourced reasoning models. For `VGS`, $\pm$ indicates standard deviation across 3 seeds, and for MV, WMV, Pass@N, it denotes bootstrap with 100 repetitions. We **bold** the highest avg. accuracy and underline second highest.

## C.2 Per-benchmark Plots for Fig. 3

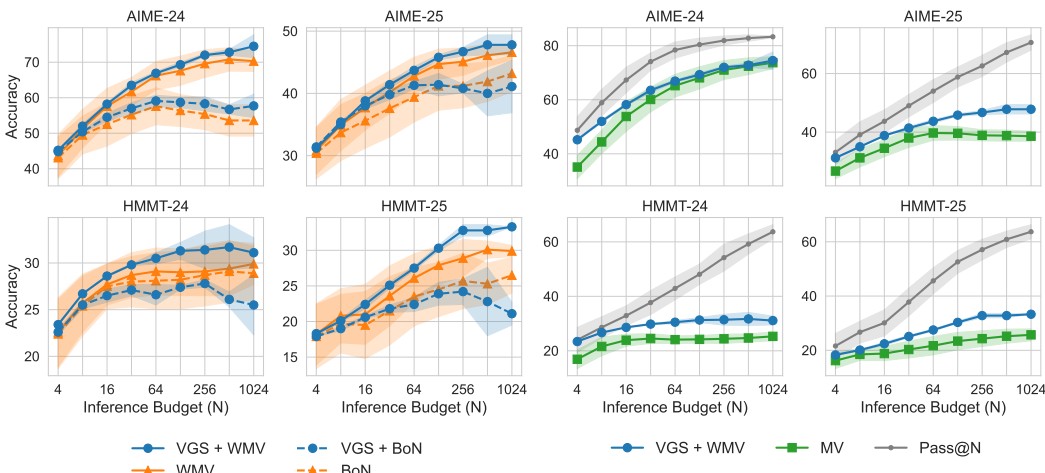

Figure 10: **Per-benchmark results for Fig. 3.** (Left) Compares best-of-$N$ (BoN), weighted majority voting (WMV) and VGS with either BoN or WMV for the final aggregation. (Right) Compares VGS to majority voting (MV), a standard baseline that does not require a scorer.

## C.3 Per-benchmark Plots for Fig. 4

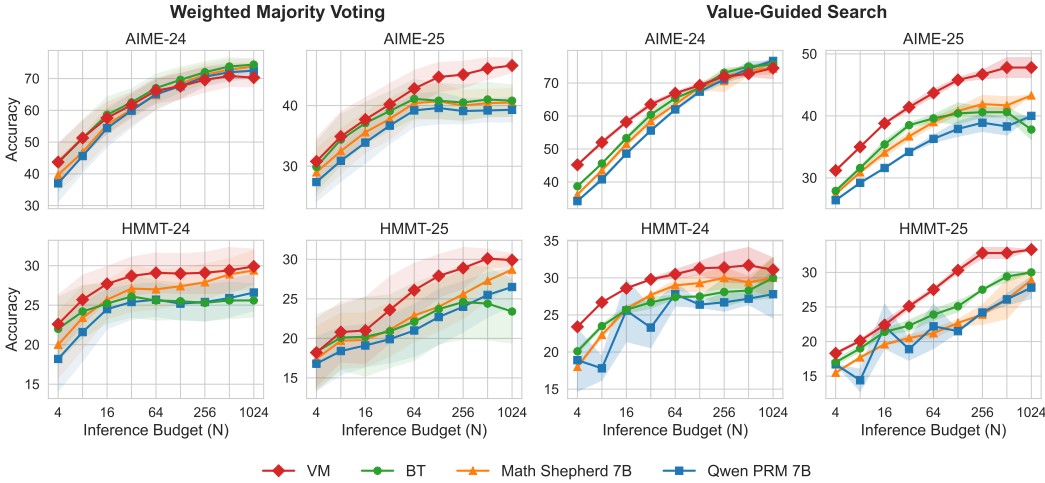

Figure 11: **Per-benchmark results for Fig. 4.** Comparison of our 1.5B value model (VM), our 1.5B Bradley-Terry reward model (BT), and two 7B state-of-the-art PRMs for two TTC scaling methods: (Left) WMV or (Right) VGS (with WMV as a final aggregation step).

## C.4 Per-benchmark Plots for Fig. 5

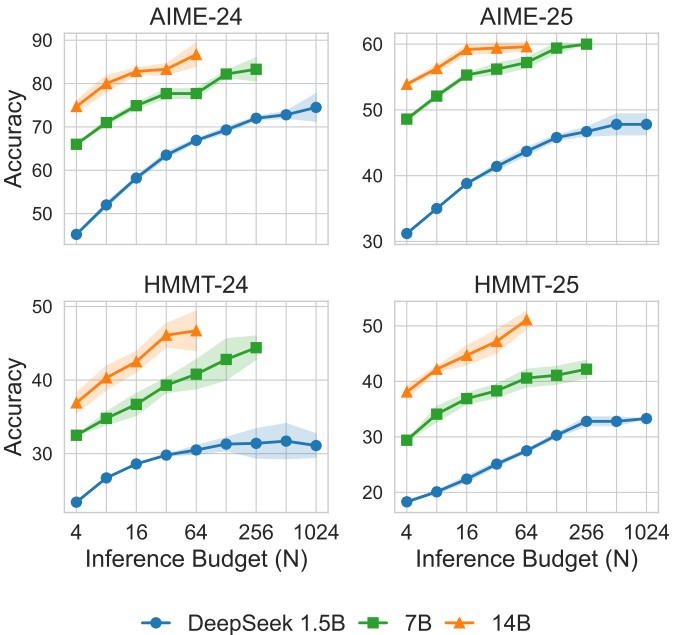

Figure 12: **Per-benchmark results for Fig. 5.** With the same `DeepSeek-VM-1.5B` providing guidance, search continues to improve with more test-time compute.

## C.5 Per-benchmark Plots for Fig. 8

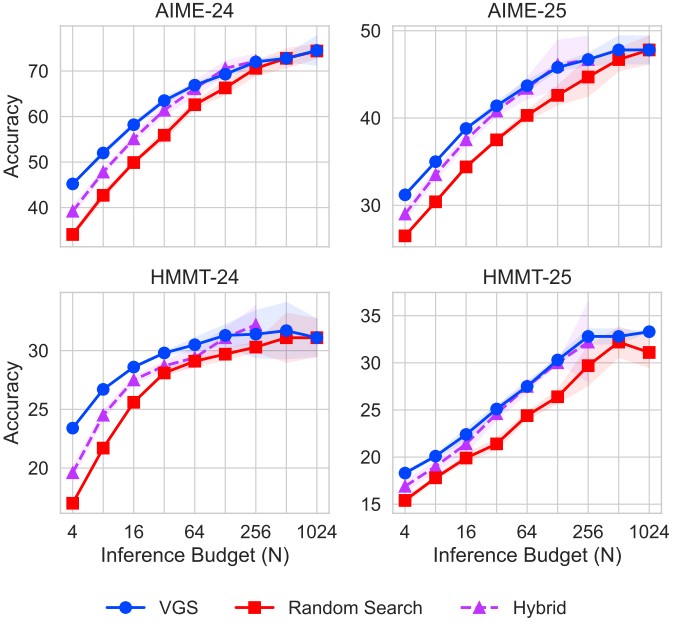

Figure 13: **Per-benchmark results for Fig. 8.** Random search is the same search process as `VGS` except intermediate blocks are randomly selected instead of using our value model.

## C.6 Results for Guiding a PPO Policy

Guo et al. [19] mentions that the performance of distilled DeepSeek models can be further enhanced through reinforcement learning (RL). In this section, we explore whether we could guide the generation of a RL trained policy. Specifically, we apply Proximal Policy Optimization (PPO) [34] to `DeepSeek-1.5B` using prompts from `OpenR1-Cleaned` and guide the trained model with `DeepSeek-VM-1.5B`.

We perform full parameter training with 8 H100 GPUs and use the same model as the policy for critic. We use a rule-based reward function based solely on the correctness of the response, assigning +1 for correct answers and 0 for incorrect or incomplete ones. To ensure that the learned policy $\pi$ remains close to the reference policy $\pi_{\text{ref}}$, an additional KL penalty is applied to the reward:

$$r(x, y) - \gamma_{\text{KL}} \left( \ln \pi(y \mid x) - \ln \pi_{\text{ref}}(y \mid x) \right), \tag{1}$$

where $r(x, y)$ is the rule-based reward for prompt $x$ and response $y$, and $\gamma_{\text{KL}}$ controls the strength of the KL penalty. To further encourage exploration, we apply standard entropy regularization by subtracting the policy entropy from the loss weighted by a coefficient $\gamma_{\text{entropy}}$:

$$\mathcal{L}_{\text{PPO}} - \gamma_{\text{entropy}} \, \mathcal{H}[\pi(\cdot \mid x)], \tag{2}$$

The hyperparameter settings are shown below.

**PPO Hyperparameter Setting**

| Setting | Parameters | |
|---|---|---|
| Generation (train) | temperature: 1.0 | top p: 1 |
| PPO | batch size: 256
mini batch size: 128
micro batch size: 1
policy learning rate: 1e-6
critic learning rate: 1e-5
train epochs: 25 | $\gamma_{\text{entropy}}$: 1e-3
$\gamma_{\text{KL}}$: 1e-4
gae $\gamma$: 1
gae $\lambda$: 1
clip ratio: 0.2
Total number of steps: 2250 |

| DeepSeek-1.5B | Pass@4 | Pass@8 | Pass@16 | Pass@32 | Pass@64 | Pass@128 | Pass@256 |
|---|---|---|---|---|---|---|---|
| AIME-24 | $48.7 \pm 5.0$ | $58.9 \pm 4.9$ | $67.3 \pm 4.7$ | $74.1 \pm 3.2$ | $78.4 \pm 3.0$ | $80.4 \pm 2.4$ | $81.9 \pm 1.7$ |
| AIME-25 | $33.1 \pm 4.1$ | $39.1 \pm 4.4$ | $43.7 \pm 3.9$ | $49.0 \pm 3.8$ | $54.0 \pm 3.9$ | $58.8 \pm 3.4$ | $62.6 \pm 3.6$ |
| HMMT-24 | $24.0 \pm 4.6$ | $28.6 \pm 3.9$ | $32.9 \pm 3.6$ | $37.7 \pm 4.4$ | $42.9 \pm 4.2$ | $48.1 \pm 3.7$ | $54.2 \pm 4.9$ |
| HMMT-25 | $21.6 \pm 4.6$ | $26.7 \pm 4.0$ | $30.1 \pm 4.8$ | $37.8 \pm 5.1$ | $45.6 \pm 4.6$ | $52.6 \pm 4.2$ | $57.1 \pm 3.8$ |

| DeepSeek-1.5B-PPO | Pass@4 | Pass@8 | Pass@16 | Pass@32 | Pass@64 | Pass@128 | Pass@256 |
|---|---|---|---|---|---|---|---|
| AIME-24 | $54.0 \pm 5.0$ | $61.4 \pm 4.6$ | $67.6 \pm 4.1$ | $73.3 \pm 3.4$ | $76.8 \pm 2.4$ | $78.3 \pm 1.7$ | $79.6 \pm 1.1$ |
| AIME-25 | $35.9 \pm 3.9$ | $39.8 \pm 3.8$ | $45.8 \pm 4.2$ | $50.9 \pm 3.5$ | $56.1 \pm 3.0$ | $59.8 \pm 3.6$ | $64.1 \pm 3.9$ |
| HMMT-24 | $27.2 \pm 4.2$ | $32.8 \pm 4.4$ | $37.6 \pm 3.6$ | $41.5 \pm 3.6$ | $45.0 \pm 3.3$ | $48.8 \pm 3.3$ | $52.4 \pm 3.4$ |
| HMMT-25 | $22.8 \pm 4.0$ | $26.8 \pm 4.3$ | $32.5 \pm 4.4$ | $36.9 \pm 4.4$ | $43.7 \pm 3.8$ | $48.9 \pm 3.8$ | $52.6 \pm 3.6$ |

Table 3: Pass@N results for `DeepSeek-1.5B` model and PPO-trained `DeepSeek-1.5B-PPO` model.

Table 3 presents the comparison between the `DeepSeek-1.5B` model and the PPO-trained model (`DeepSeek-1.5B-PPO`). As N increases, the performance gap gradually narrows. While the PPO-trained model performs competitively at lower $N$ values, it is surpassed by the base model at Pass@32 on both the AIME-24 and HMMT-25 datasets. This decline in performance could be attributed to the reduced entropy of the model after PPO training, which limits the diversity of model generations and negatively impacts performance at higher Pass@N.

We report our value-guided results of the PPO-trained model in Fig. 14. We observe that VGS nicely complements PPO training and provides additional test-time compute gains in performance compared to WMV and MV.

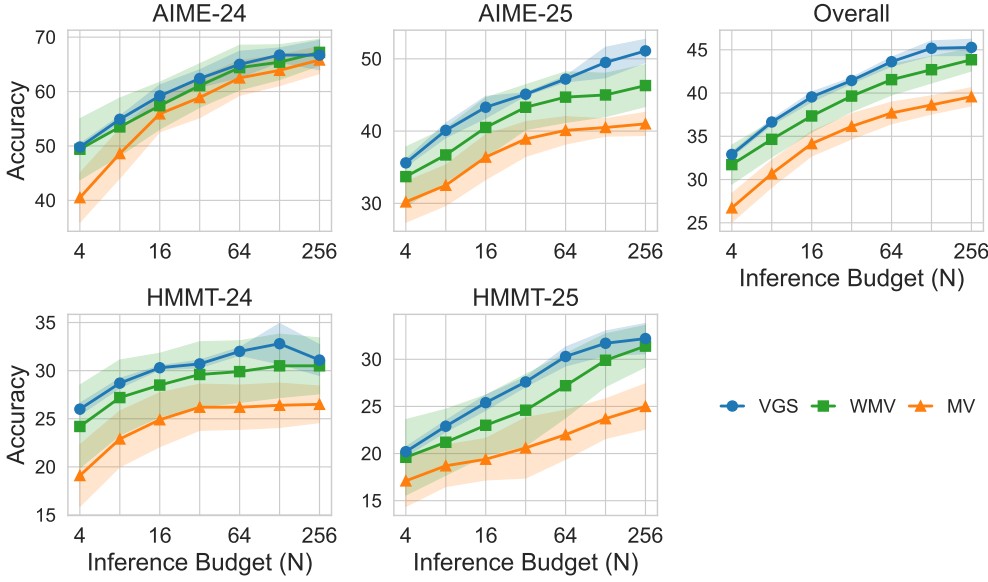

Figure 14: **Guiding DeepSeek-1.5B Trained with PPO.** Comparison of VGS, WMV and MV for TTC scaling our PPO policy.

## C.7 Qualitative Examples

Figures 17, 18, and 19 (at the end of the paper) show representative qualitative examples where value scores from `VGS` are used to guide beam search. At each step, two blocks of tokens are proposed, and the one with the higher value is selected to continue the solution. Due to space constraints, parts of the beams are abridged with . . . , and for ease of visualization, blue highlights indicate correct reasoning steps, while red highlights denote incorrect ones.

Low-scoring beams exhibit different types of failure. In Figure 17, the rejected beam alternates between correct and incorrect steps, resulting in confused and ultimately incorrect reasoning. In Figure 18, the beam begins with a plausible strategy involving GCD analysis but eventually resorts to ineffective trial and error. In Figure 19, the beam makes a critical error in the algebraic transformation early on and fails to recover. In contrast, the selected beams across all examples demonstrate systematic reasoning and successfully solve the problems.

Interestingly, despite the critical error in Figure 19, `VGS` assigns a moderately high score (0.337) to the rejected beam—higher than scores for less subtle failures in earlier examples—suggesting that even significant mistakes can be difficult to detect when embedded in otherwise coherent reasoning.

Finally, we empirically compare the distribution of generation lengths between the `DeepSeek-1.5B` base model and `VGS` with `DeepSeek-1.5B` across all benchmarks (Figure 15). On average, `VGS` generates noticeably shorter responses (11,219 tokens vs. 12,793 for the `DeepSeek-1.5B` base model), suggesting that beam search not only enhances accuracy but also promotes more concise reasoning. This trend is consistent with our qualitative analysis, where beam search tends to favor token blocks that are direct and solution-oriented, rather than verbose or meandering reasoning. Notably, the sharp peak near 16,000 tokens corresponds to the maximum generation length of DeepSeek models (16,384). For the base model, as many as 50% of the generations reach this limit, often resulting in incomplete outputs.

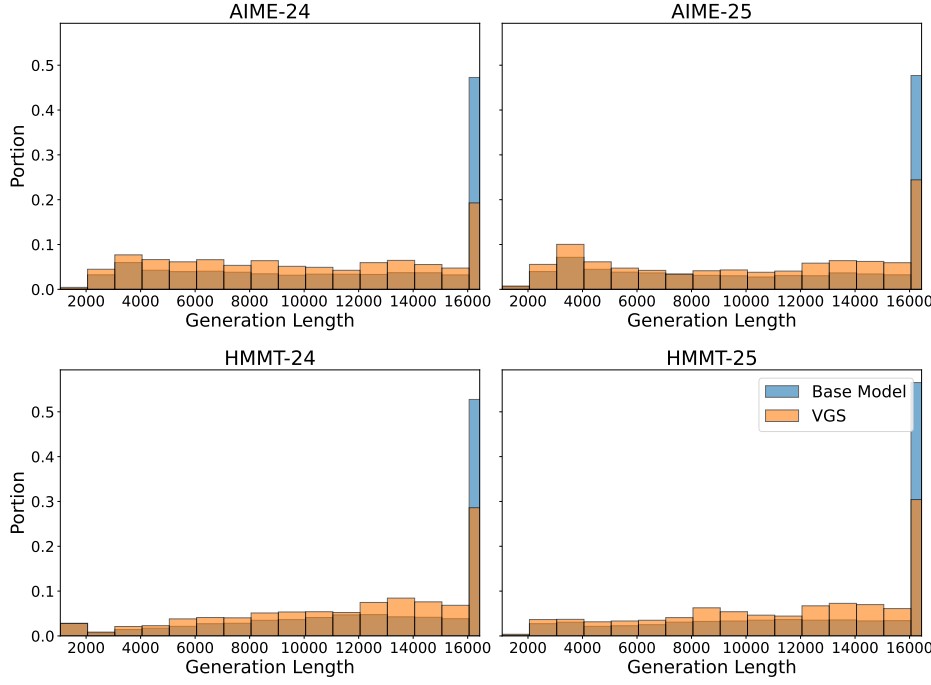

Figure 15: **Histogram of generation lengths for the** `DeepSeek-1.5B` **base model vs.** `VGS`. `VGS` consistently produces shorter responses across benchmarks, with average lengths of 11,219 and 12,793 tokens for `VGS` and the base model, respectively. The peak around 16,000 tokens reflects the generation cap of DeepSeek models, which the base model frequently hits, often resulting in incomplete outputs.

# D   Further Details of Data Collection

**Pre-filtering process.** Here we describe the pre-filtering process for constructing `OpenR1-Cleaned` in more detail. Below are the sequence of filtering operations we performed on `OpenR1-Math` [2]. We arrived at these rules by manually inspecting the data, by sampling 100 random problems from the dataset and checking if all problems' solutions looked reasonable to us. In `OpenR1-Math`, a solution is a fleshed-out solution to the math problem, an answer is the final answer to the math problem.

1. Filter out all solutions with 0 or $> 1$ boxed answers (enclosed in \boxed{}). These are ambiguous and difficult to parse out the answer.
2. Filter out answers which are difficult to automatically parse or verify. This includes answers containing: 'or', 'and' \mathrm, \quad, answers with equal signs, commas, semicolons, \cup, \cap, inequality symbols, approximation symbols.
3. Filter out multiple-choice questions, which are labeled with `question_type` = 'MCQ'.
4. Filter out questions with multiple parts, as it is ambiguous which part the answer is for.
5. Filter out questions containing links (`http://` or `https://`), since the models we test cannot access the web.

**Roll-in roll-out process.** We also provide further intuition for the roll-in vs. roll-out process (illustrated in Fig. 2 left). The roll-in and then roll-out process is a standard technique in imitation learning [12] and reinforcement learning [11].

**Roll-out.** The roll-out process uses a fixed policy $\pi^{\text{ref}}$ to roll-out from any partial solution provided by the roll-in process. The rationale for using a fixed roll-out policy is to fix the target of the classification / value regression problem. In particular, the classifier is trained to predict the probability of each class under the roll-out policy, given the partial solution.

**Roll-in.** The main point of the roll-in process is to create a diverse distribution of partial solutions to roll-out from. By creating a diverse roll-in distribution with multiple roll-in policies, we can ensure that the classifier is trained on a diverse context distribution and will generalize better to new traces. To select where to cut a roll-in (to start the roll-out), we sample a cut index from the distribution of $p(i) = \frac{\sqrt{i}}{\sum_{j \in [L]} \sqrt{j}}$ where $L$ is the length of the roll-in. We chose this such that the cut index is more likely to occur at earlier positions of the roll-in. We want to encourage more learning at earlier positions since those prediction problems are more difficult than at later positions. The following figure (Fig. 16) illustrates the distribution of the length of roll-outs, which shows that this indexing scheme indeed yields many long roll-outs.

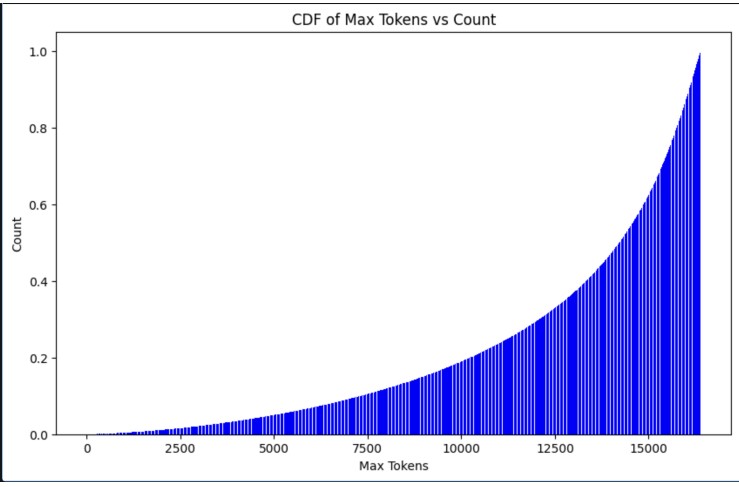

Figure 16: **Distribution of roll-out length.**

# E  Further Details for Value Model Training

Our value model `DeepSeek-VM-1.5B` uses the same base architecture as the `DeepSeek-R1-Distill-Qwen-1.5B` model, which is a 1.5B parameter transformer model with 28 layers and 1536 hidden size. To turn this model into a value model, we replace the LM head with a scoring head, parameterized by a two-layer MLP, that outputs logits for three classes: 'correct', 'incorrect', and 'incomplete'. The number of classes can be modified to suit the task at hand.

| Category | Parameter | Value |
|---|---|---|
| Model | Base Model Initialization | `DeepSeek-R1-Distill-Qwen-1.5B` |
| | Hidden size $d_{\text{model}}$ | 1536 |
| | Score Head | two-layer MLP with hidden size $d_{\text{model}}$ |
| | Score Head Bias | False |
| | Score Head Labels | 0=Incorrect, 1=Correct, 2=Incomplete |
| Data | Dataset | `OpenR1-VM` |
| | Validation split | 500 |
| | Max sequence length | 16384 |
| Training | Batch size | 1024 |
| | Learning-rate schedule | Cosine with `max_lr=1e-4` |
| | Warm-up steps | 10% of total steps |
| | Dropout | 0.05 |
| | Number of epochs | 5 |
| Optimizer | Optimizer type | AdamW |
| | $\beta_1$ | 0.9 |
| | $\beta_2$ | 0.95 |
| | Weight decay | 0.1 |
| | Grad Norm Clip | 5 |
| Compute | GPUs | 16 nodes of $8 \times$ `NVIDIA H100` |
| | Wall-clock time (h) | 24 hours |
| | Tokens Throughput (tokens/s) | 2.07M |
| | Loss Tokens Throughput (loss tokens/s) | 835k |
| | Total Tokens Processed (per epoch) | 35.7B |
| | Total Loss Tokens Processed (per epoch) | 14.4B |

Table 4: Value Model Training Parameters.

We sweeped learning rates 1e-4, 3e-4, 7e-5 and we save checkpoints at every epoch. We selected the best checkpoint via WMV performance on AIME-24.

## F  Further Details for Inference with Search

Given a `problem`, the prompt we used is:

```
<|begin_of_sentence|><|User|>{problem} Please think
step-by-step and put your final answer within
\boxed{}.<|Assistant|><think>\n
```

We use the same decoding parameters as in the original DeepSeek paper [19].

| Category | Parameter | Value |
|---|---|---|
| Decoding | Inference Engine | SGLang [52] |
| | Max generation length | 16384 |
| | Temperature | 0.6 |
| | Top-p | 0.95 |
| | Think Token | `<think>` |
| | End of Think Token | `</think>` |
| Best Parameters for Search (`VGS`) | Model | `DeepSeek-VM-1.5B` |
| | Beam width | 2 |
| | Block size (tokens) | 4096 |
| | Parallel branches (DVTS) | budget dependent |
| | Final aggregation rule | Weighted Majority Vote (WMV) |

Table 5: Decoding and Search Parameters.

## G  Further Details for Training Bradley-Terry Reward Model

**Dataset.** Recall that our dataset for value model training (`OpenR1-VM`) contains $56$ responses per problem. To construct a Bradley-Terry dataset, we sample up to $4$ response pairs per problem, where a pair consists of a response with reward $0$ (the 'reject' response) and a response with reward $1$ (the 'chosen' response). Some prompts may have fewer than $4$ responses with reward $0$ or $1$, and in those cases, we include as many as possible. This yields a dataset of roughly 122k pairs.

**Model.** We use the same model architecture as the value model (Appendix E) except that the score head outputs a single scalar score instead of a three-dimensional vector. We use the same training pipeline, except that the training loss is swapped to the standard BT loss [8]:

$$L_{\text{BT}}(\theta, \mathcal{B}) = \frac{1}{|\mathcal{B}|} \sum_{(x, y_r, y_c) \in \mathcal{B}} - \log \sigma(f_\theta(x, y_c) - f_\theta(x, y_r)), \tag{3}$$

where $\sigma$ is the sigmoid function, $y_r$ is the reject response and $y_c$ is the chosen response. We use a batch size of 128 pairs and we train for one epoch. We sweeped many learning rates: 3e-4, 1e-4, 7e-5, 3e-5, 7e-6. We found that the BT loss drops and plateaus much quicker than the value model loss, and all learning rates yielded similar final losses. We consider the last ckpt of each run and we selected `lr=3e-5` as best ckpt with search (width 2, block size 4096, WMV aggregation) on `aime-24`. We note that one detail for getting BT to work with weighted majority is to use the sigmoid of the BT score, *i.e.*, take WMV with $\sigma(f_\theta(x, y))$ instead of $f_\theta(x, y)$ itself. While this doesn't affect BoN performance, we found that taking sigmoid was crucial for WMV performance to scale well.

# H   Inference FLOPS Computation

In this section, we compute the FLOPs for search and show that adding value model guidance at the block-level introduces negligible compute overhead, as the vast majority of the compute is from generator model. We follow the approach outlined in Kaplan et al. [22], Sardana et al. [33] to compute the FLOPs for a single forward pass of the transformer, ignoring the embedding layers for simplicity. Consider a transformer with $n_{layer}$ layers, $d$ dimensional residual stream, $d_{ff}$ dimensional feed-forward layer, and $d$ dimensional attention outputs. Then the number of non-embedding parameters is $N = 2n_{layer}d(2d + d_{ff})$, and the number of FLOPs for a single forward pass over a context of length $n_{ctx}$ is

$$C(n_{ctx}) = 2N + 2n_{layer}n_{ctx}d. \tag{4}$$

Then, in the regime where $d_{model} > n_{ctx}/12$, Kaplan et al. [22], Sardana et al. [33] further approximate the above by ignoring the $n_{ctx}$ term, *i.e.*, $C$ becomes independent of $n_{ctx}$. We adopt this approximation when estimating the inference FLOPs of our generator models. Thus, for a context length of $n_{ctx} = 16,384$, the inference FLOPs for one complete generation for each generator model is $2Nn_{ctx}$:

1. `DeepSeek-R1-Distill-Qwen-1.5B`: $2 \times 1.5B \times 16384 = 49.1T$.
2. `DeepSeek-R1-Distill-Qwen-7B`: $2 \times 7B \times 16384 = 229T$.
3. `DeepSeek-R1-Distill-Qwen-14B`: $2 \times 14B \times 16384 = 459T$.
4. `DeepSeek-R1` (671B, with 37B activated params): $2 \times 37B \times 16384 = 1212T$.

We now compute the FLOPs needed for one forward pass of the value model. Since we use a block-size of 4096, there are at most $16384/4096 = 4$ value model inferences per generation. Thus, the FLOPs from the value model is:

1. 1.5B classifier: $2 \times 1.5B \times 4 = 12B$.
2. 7B classifier (baselines): $2 \times 7B \times 4 = 56B$.

Thus, we can see that the value model FLOPs is negligible compared to the generator model FLOPs. In particular, when guiding a 1.5B generator with a 1.5B classifier, the classifier FLOPs is only $0.024\%$ of the generator FLOPs. With a compute budget of 256, this amounts to a total FLOPs of $(49.1T + 12B) \times 256 = 12.6P$. When guiding with a 7B classifier, the total FLOPs is $(49.1T + 56B) \times 256 = 12.6P$. Note that the FLOPs required for generating 256 independent generations is $49.1T \times 256 = 12.6P$. Thus, search has a negligible overhead comapred to (weighted) majority voting or best of $n$.

**AIME-24 Problem:** Jen enters a lottery by picking 4 distinct numbers from $S = \{1, 2, 3, \cdots, 9, 10\}$. 4 numbers are randomly chosen from $S$. She wins a prize if at least two of her numbers were 2 of the randomly chosen numbers, and wins the grand prize if all four of her numbers were the randomly chosen numbers. The probability of her winning the grand prize given that she won a prize is $\frac{m}{n}$ where $m$ and $n$ are relatively prime positive integers. Find $m + n$.

- - - - - - - - - - - - - - - - - - - - - - - - - - - - - - - - - - - - - - - - - - - - - - - - - - - - - - - - - -

**Rejected beam with score 0.009:** Okay, so I need to figure out the probability that Jen wins the grand prize given that she won a prize. That means I have to use conditional probability. I remember that conditional probability is $P(A|B) = P(A \cap B)/P(B)$. In this case, event A is her winning the grand prize, and event B is her winning a prize. So, I need to find the probability that she won both the grand prize and a prize, which is just her winning the grand prize, divided by the probability that she won a prize. First, let me figure out the total number of possible ways Jen can pick 4 numbers from S, which has 10 numbers. That should be the combination of 10 choose 4, right? So, $C(10, 4) = 210$. So, the total number of possible combinations is 210.

So, she can win a prize in two scenarios: either exactly two of her numbers are the randomly chosen ones, or all four are. . . . The number of ways she can have exactly two numbers correct is $C(4, 2)$ ways to choose the correct numbers, and then $C(6, 2)$ ways to choose the incorrect ones from the remaining 6 numbers. Similarly, the number of ways she can have all four numbers correct is just $C(4, 4) = 1$. So, the total number of winning ways is $C(4, 2) * C(6, 2) + C(4, 4)$. . . . $C(4, 2)$ is 6, and $C(6, 2)$ is 15. So, 6*15 = 90. Then, $C(4, 4)$ is 1. So, total winning ways are 90 + 1 = 91.

Alternatively, I can compute the probability of not winning a prize, which is the probability that she has 0 or 1 correct numbers. Let me compute that and subtract from 1 to see if it matches. Number of ways to have 0 correct numbers: $C(4, 0) * C(6, 4) = 1 * 15 = 15$. Number of ways to have 1 correct number: $C(4, 1) * C(6, 3) = 4 * 20 = 80$. So, total non-winning ways: 15 + 80 = 95. Therefore, probability of not winning a prize is 95/210, so probability of winning a prize is $1 - 95/210 = (210 - 95)/210 = 115/210$. Wait, that doesn't match with the previous result. Hmm, something's wrong here.

Wait, why did I get 91 before? Let me see. I had $C(4, 2) * C(6, 2) + C(4, 4)$. So, that's 6*15 +1=91. But that's only the number of winning ways where she has exactly two or exactly four correct numbers. So, 91 winning ways, 95 non-winning, which adds to 186, which is less than 210. So, the rest must be the impossible case? No, because in reality, she can't have 0 correct numbers if she has 4 numbers. Wait, no, she can have 0 correct numbers if all her numbers are among the 6 incorrect ones. So, that's 15 ways. Similarly, 1 correct number is 80 ways, and 2 correct numbers is 90 ways, and 4 correct numbers is 1 way. So, 15 +80 +90 +1=186. Wait, so actually, 186 winning ways and 24 non-winning ways. So, probability of winning is 186/210, which simplifies to 31/35. . . .

Therefore, $P(A|B) = (1/210)/(186/210) = 1/186$. So, m is 1 and n is 186. Are they relatively prime? 1 and 186 share no common factors besides 1, so yes. So, $m + n = 1 + 186 = 187$. . . .

- - - - - - - - - - - - - - - - - - - - - - - - - - - - - - - - - - - - - - - - - - - - - - - - - - - - - - - - - -

**Selected Beam with score 0.996:** Okay, so this is a conditional probability problem. The formula for conditional probability is $P(A|B) = P(A \cap B)/P(B)$, where A is the event that she wins the grand prize, and B is the event that she wins a prize. Since winning the grand prize is a specific case of winning a prize, A is a subset of B. So, $P(A \cap B)$ is just $P(A)$. Therefore, the probability we're looking for is $P(A)/P(B)$. . . .

Since she picks 4 numbers from 10, the total number of possible 4-number combinations is $C(10, 4)$. The number of favorable outcomes for A is 1 (since there's only one exact set of four numbers that could be the randomly chosen ones). So, $P(A)$ is $1/C(10, 4)$. . . .

To find $P(B)$, it might be easier to compute the probability of the complementary event, which is that Jen wins zero prizes or one prize, and subtract that from 1. So, $P(B) = 1$ - P(0 prizes) - P(1 prize). Let me compute P(0 prizes) and P(1 prize).

First, P(0 prizes): This is the probability that none of Jen's four numbers are among the four randomly chosen. So, all four numbers she picked are from the remaining six numbers (since S has 10 numbers, and if she picks 4, the remaining are 6). So, the number of ways to choose 4 numbers from 6 is $C(6, 4)$. Therefore, P(0 prizes) = $C(6, 4)/C(10, 4)$.

Similarly, P(1 prize): This is the probability that exactly one of Jen's numbers is among the four randomly chosen. So, we need to choose 1 number from her 4, and 3 numbers from the remaining 6 (since 10 - 4 = 6). So, the number of ways is $C(4, 1) * C(6, 3)$. Therefore, P(1 prize) = $[C(4, 1) * C(6, 3)]/C(10, 4)$. . . . So, putting it all together: $P(B) = 1 - [C(6, 4)/C(10, 4)] - [C(4, 1) * C(6, 3)/C(10, 4)]$ . . .

Therefore, $P(B) = 1 - 15/210 - 80/210$. . . . Therefore, $P(B)$ is 23/42. . . . Therefore, the conditional probability $P(A|B)$ is $(1/210)/(23/42) = (1/210) * (42/23) = (42)/(210 * 23)$ . . . Therefore, the probability becomes $(1/5)/23 = 1/(5 * 23) = 1/115$. . . . But 1/115 can't be simplified further, so m is 1 and n is 115, which are coprime. Therefore, m + n is 1 + 115 = 116. . . .

Figure 17: Example of selected and rejected beams during beam search with VGS. The high-scoring beam (score 0.996) follows a correct and coherent line of reasoning, arriving at the correct answer. In contrast, the rejected beam (score 0.009) contains several inconsistencies and incorrect steps, despite occasionally making correct logical deduction—demonstrating the effectiveness of VGS as a value model. Highlighting is added for clarity: blue indicates correct reasoning steps or results, while red indicates incorrect ones. Part of generations are abridged with . . . notation.

**AIME-25 Problem:** Find the sum of all positive integers $n$ such that $n + 2$ divides the product $3(n+3)(n^2 + 9)$.

---

**Rejected beam with score 0.395:** First, let's parse the problem. I need to find all positive integers $n$ where $n + 2$ is a divisor of $3(n+3)(n^2 + 9)$. So, mathematically, this means that $n + 2$ divides $3(n+3)(n^2 + 9)$. In other words, when you divide $3(n+3)(n^2 + 9)$ by $n + 2$, there should be no remainder. So, $n + 2$ must be a factor of this product.

One way to approach this is by using the Factor Theorem. The Factor Theorem states that if $n + 2$ is a factor of a polynomial $P(n)$, then $P(-2) = 0$. So, if I let $P(n) = 3(n+3)(n^2 + 9)$, then $P(-2)$ should be equal to zero if $n + 2$ divides $P(n)$. Let me compute $P(-2)$:

$P(-2) = 3(-2 + 3)((-2)^2 + 9)$ Simplify each part: $(-2 + 3) = 1$ $(-2)^2 = 4$, so $4 + 9 = 13$ Thus, $P(-2) = 3 * 1 * 13 = 39 \ldots$ Wait, perhaps I'm misapplying the Factor Theorem here. Let me think again.

Alternatively, another approach is to express $n + 2$ as $k$, so $k = n + 2$. Then, $n = k - 2$. Substitute $n = k - 2$ into the expression $3(n+3)(n^2 + 9)$ and see what conditions must hold for $k$ to divide this. $\ldots$ Therefore, the product becomes: $3(k + 1)(k^2 - 4k + 13) \ldots$

Which implies that $k$ divides $3(k + 1)(k^2 - 4k + 13)$. Since $k$ divides the product, it must divide at least one of the factors. But $k$ divides $k$, so we can consider the other factors.

But wait, $k$ divides $3(k + 1)(k^2 - 4k + 13)$, so $k$ divides $3(k + 1)(k^2 - 4k + 13)$. Since $k$ and $k + 1$ are consecutive integers, their GCD is 1. Similarly, $k$ and $k^2 - 4k + 13$ may have a GCD. Let me check:

Let me denote $d = \gcd(k, k^2 - 4k + 13)$. Then $d$ divides $k$ and $k^2 - 4k + 13$. So, $d$ divides $k^2 - 4k + 13 - k(k - 4) = k^2 - 4k + 13 - k^2 + 4k = 13$. So, $d$ divides 13. Therefore, $d = 1$ or 13. Wait, maybe I'm complicating things. Let me approach this differently. $\ldots$

But maybe there's another way to approach this. Let me try plugging in small positive integers for $n$ and see if $n + 2$ divides the product $3(n+3)(n^2 + 9)$. That might help confirm if $n = 1$ is the only solution.

Testing $n = 1$: As above, product is 120, which is divisible by 3.
Testing $n = 2$: $n + 2 = 4$. Product is $3(5)(13) = 195$. 195 divided by 4 is 48.75, which isn't an integer. So, 4 doesn't divide 195.
$\ldots$
Testing $n = 9$: $n + 2 = 11$. Product is $3(12)(90) = 3240$. 3240 divided by 11 is approximately 294.545, not integer.
Testing $n = 10$: $n + 2 = 12$. Product is $3(13)(109) = 4209$. 4209 divided by 12 is 350.75, not integer.
Wait, so up to $n = 10$, only $n = 1$ works. Maybe $n = 1$ is the only solution. $\ldots$

---

**Selected Beam with score 0.998:** First, let me understand what the problem is asking. It says that $n + 2$ divides $3(n+3)(n^2 + 9)$. In other words, when we divide the product $3(n+3)(n^2 + 9)$ by $n + 2$, the result should be an integer. So, I need to find all positive integers $n$ for which this condition holds and then sum them up.

Let me write this out mathematically. If $n + 2$ divides $3(n+3)(n^2 + 9)$, then there exists some integer $k$ such that:

$$3(n+3)(n^2 + 9) = k(n + 2)$$

But since we're looking for integer solutions, maybe it's better to think about this in terms of divisibility. So, $n + 2$ must be a divisor of $3(n+3)(n^2 + 9)$. That suggests that $n + 2$ divides this product.

I remember that if a number $a$ divides another number $b$, then $b$ modulo $a$ is zero. So, maybe I can use modular arithmetic here. Let me see.

Since $n + 2$ divides $3(n+3)(n^2 + 9)$, it must be that:

$$3(n+3)(n^2 + 9) \equiv 0 \pmod{n + 2}$$

$\ldots$ First, $n + 3$ modulo $n + 2$. Let's substitute $n = -2 + k$, but maybe that's not necessary. Alternatively, we can note that $n + 3 = (n + 2) + 1$. So, modulo $n + 2$, $n + 3 \equiv 1$.

Similarly, $n^2 + 9$. Let me compute $n^2$ modulo $n + 2$. Since $n \equiv -2 \pmod{n + 2}$, then $n^2 \equiv (-2)^2 = 4 \pmod{n + 2}$. Therefore, $n^2 + 9 \equiv 4 + 9 = 13 \pmod{n + 2}$.

So, putting it all together, modulo $n + 2$, the product becomes:

$$3(n+3)(n^2 + 9) \equiv 3 \times 1 \times 13 = 39 \pmod{n + 2}$$

So, this implies that $39 \equiv 0 \pmod{n + 2}$, which means that $n + 2$ divides 39. Therefore, $n + 2$ must be a positive divisor of 39. Now, the positive divisors of 39 are 1, 3, 13, and 39. But since $n$ is a positive integer, $n + 2$ must be at least 3 (because if $n = 1$, $n + 2 = 3$). So, the possible values for $n + 2$ are 3, 13, and 39. Therefore, the corresponding values for $n$ are: - If $n + 2 = 3$, then $n = 1$. - If $n + 2 = 13$, then $n = 11$. - If $n + 2 = 39$, then $n = 37$. $\ldots$ Therefore, the sum of all such positive integers $n$ is $1 + 11 + 37$. Let me compute that: $1 + 11$ is 12, $12 + 37$ is 49. $\ldots$

Figure 18: Additional examples of selected and rejected beams under VGS. The high-scoring beam applies a concise and effective modular reasoning strategy whereas the rejected beam attempts several approaches—including an incorrect use of the Factor Theorem and trial-and-error—but ultimately fails. Color coding follows Figure 17. Part of generations are abridged with . . . notation.

**HMMT-25 Problem:** Mark writes the expression $\sqrt{\underline{abcd}}$ on the board, where $\underline{abcd}$ is a four-digit number and $a \neq 0$. Derek, a toddler, decides to move the $a$, changing Mark's expression to $a\sqrt{\underline{bcd}}$. Surprisingly, these two expressions are equal. Compute the only possible four-digit number $\underline{abcd}$.

- - - - - - - - - - - - - - - - - - - - - - - - - - - - - - - - - - - - - - - - - - - - - - -

**Rejected beam with score 0.337:** Okay, so I have this problem where Mark writes the square root of a four-digit number, which is written as $\sqrt{(abcd)}$, where a isn't zero. Then Derek moves the first digit 'a' to the end, making the expression $a\sqrt{(bcd)}$. And the interesting part is that both expressions are equal. I need to find the only possible four-digit number $abcd$ that satisfies this condition.

First, let me try to understand the problem step by step. Mark's expression is the square root of a four-digit number, so that's $\sqrt{(1000a + 100b + 10c + d)}$. Derek changes this by moving the first digit 'a' to the end, so the new number is $1000b + 100c + 10d + a$, and Derek's expression is a multiplied by the square root of this new number, which is $a\sqrt{(1000b + 100c + 10d + a)}$. And both of these are equal.

So, mathematically, I can write this as:

$\sqrt{(1000a + 100b + 10c + d)} = a\sqrt{(1000b + 100c + 10d + a)}$

If I square both sides to eliminate the square roots, I get:

So, $1000a + 100b + 10c + d = a * (1000b + 100c + 10d + a) \dots$

- - - - - - - - - - - - - - - - - - - - - - - - - - - - - - - - - - - - - - - - - - - - - - -

**Selected Beam with score 0.999:** Let me write this down step by step. Let's denote the four-digit number as $N = 1000a + 100b + 10c + d$. So, Mark's expression is $\sqrt{N}$. Then, Derek changes it to $a\sqrt{(bcd)}$, which is $a\sqrt{(100b + 10c + d)}$. According to the problem, these two expressions are equal, Which means:

$\sqrt{(1000a + 100b + 10c + d)} = a\sqrt{(100b + 10c + d)}$

Let me square both sides to eliminate the square roots:

1000a + 100b + 10c + d = a² * (100b + 10c + d)

Hmm, that gives me an equation:

1000a + 100b + 10c + d = a² (100b + 10c + d)

Let me denote the four-digit number as N, so N = 1000a + 100b + 10c + d, and the three-digit number as M = 100b + 10c + d. So, the equation becomes: N = a² * M

But since N is a four-digit number and a is a single digit (from 1 to 9, since $a \neq 0$), and M is a three-digit number (from 0 to 999). So, let me rewrite N as 1000a + M.

So, 1000a + M = a² * M

Let me rearrange this equation:

1000a = a² * M - M

Factor out M on the right side:

1000a = M(a² - 1)

Therefore, M = (1000a) / (a² - 1)

Since M must be an integer (as it's a three-digit number), (1000a) must be divisible by (a² - 1). So, (a² - 1) must be a divisor of 1000a.

Let me note that a is a digit from 1 to 9. So, I can try each a from 1 to 9 and see if (a² - 1) divides 1000a, and then check if M is a three-digit integer.

Let's compute for each a:

Starting with a=1:

a=1: M = 1000*1 / (1 - 1) = 1000 / 0, which is undefined. So, a=1 is invalid.

a=2:

M = 1000*2 / (4 - 1) = 2000 / 3 ≈ 666.666..., which is not an integer. So, a=2 is invalid.

a=3:

M = 1000*3 / (9 - 1) = 3000 / 8 = 375. That's an integer. So, M=375.

a=4:

M = 1000*4 / (16 - 1) = 4000 / 15 ≈ 266.666..., not integer. So, a=4 invalid.

a=5:

M = 1000*5 / (25 - 1) = 5000 / 24 ≈ 208.333..., not integer.

a=6:

M = 1000*6 / (36 - 1) = 6000 / 35 ≈ 171.428..., not integer.

a=7:

M = 1000*7 / (49 - 1) = 7000 / 48 ≈ 145.833..., not integer.

a=8:

M = 1000*8 / (64 - 1) = 8000 / 63 ≈ 127.450..., not integer.

a=9:

M = 1000*9 / (81 - 1) = 9000 / 80 = 112.5, which is not integer.

So, only a=3 gives us an integer M=375, which is a three-digit number. Therefore, the only possible four-digit number is 3375. $\dots$

Figure 19: Additional examples of selected and rejected beams under VGS. The rejected beam misrepresents Derek's transformation, leading to an incorrect equation that derails the solution. In contrast, the high-scoring beam correctly models the relationship and systematically solves for the unique valid value of $a$. Color coding follows Figure 17. Part of generations are abridged with $\dots$ notation.

