# OpenReview forum: "Value-Guided Search for Efficient Chain-of-Thought Reasoning"
_NeurIPS.cc/2025/Conference — NeurIPS 2025 poster_

### Official Review · Reviewer_YPfy · 2025-06-13

**Clarity:** 4
**Significance:** 3
**Originality:** 2
**Rating:** 5
**Confidence:** 2

**Summary:**

The paper proposes a new method to train a value model which does not rely on a notion of "step" by training a correctness classifier on data of the form (prompt, roll-in, roll-out) generated from DeepSeek reasoning responses for competition math problems. The value model is also applied to guide test-time search (VGS) in a blockwise manner, i.e., the block with highest value when concatenated is selected according to BFS, beam search, etc. VGS is shown to outperform existing inference compute methods including BoN, majority vote and PRM-guided search.

**Questions:**

See Weaknesses

**Ethical Concerns:**

["NO or VERY MINOR ethics concerns only"]

**Final Justification:**

The author rebuttal and other reviews have stregthened my positive assessment of the paper’s contributions towards utilizing test time compute for reasoning tasks requiring long CoT.

**Limitations:**

Yes

**Quality:**

3

**Strengths And Weaknesses:**

**Strengths**

The proposed value-guided search appears to be novel, effective and scalable compared to costly PRM-based search methods. VGS improves on both accuracy and FLOP savings for fixed model size and performance budget. The experiments and results are clearly presented and most include estimates on error bars.

**Weaknesses/Questions**

* How can VGS overcome the issue of credit assignment given that there are no explicit steps to score? How strong is the empirical correlation between block-level value scores and eventual correctness?
* For the competition math problem sets, how can data contamination be controlled for?
* The dataset generation is directly dependent on the base reasoning model. Have other base models been tried and how might they affect the resulting value model?
* It would be interesting to see how VGS performs on cross-domain tasks.

---

> ### Author Rebuttal · Authors · 2025-07-26
>
> Dear Reviewer YPfy,
>
> Thank you for your positive review and for your support of our work! We are happy to provide detailed answers to your questions below, and we hope they will further increase your confidence in your assessment.
>
> ### 1. On Credit Assignment
> Thanks for this great question. You are correct that there are no explicit steps to score since we train a token-level value model, which operates at the same granularity as the autoregressive LM. To see the benefit of selecting intermediate blocks with our value model, we performed the random search ablation presented in Figure 8. The results show that selecting blocks with a high value score leads to significantly improved final performance compared to random selection. This provides strong empirical evidence for a positive correlation between higher intermediate value scores and more promising reasoning blocks.
>
> Moreover, we remark that despite having sparse rewards, our value model estimates the expected return at any partial response, and thus can still provide dense signals about the likelihood of this state leading to high reward. In other words, even when rewards are sparse, there are partial responses and blocks associated with non-zero rewards. We note that our approach is philosophically aligned with Sutton's "bitter lesson," favoring general methods that scale with data and compute over those relying on human-defined structures such as a "step." Indeed, our VGS method using a 1.5B value model, empirically outperforms a 7B PRM baseline trained with step-level feedback (Table 1, VGS w/ DeepSeek-VM-1.5B has 45.7% accuracy vs. VGS w/ Qwen2.5-Math-PRM-7B has 40.2% accuracy). Ultimately, we hypothesize it may be useful to learn an appropriate notion of "step" beyond the existing heuristics (i.e. separating by newline), but we leave that as promising future work.
>
> ### 2. On Controlling for Data Contamination
> Thanks for this question. To ensure there is no data contamination, we will focus our evaluations on only AIME-25 and HMMT-25, which are math competitions that happened after the release dates of DeepSeek and OpenR1 (our dataset is a subset of OpenR1). This certainly controls for data contamination because we only evaluate on problems after the release dates of both the underlying model and the dataset.
>
> For example, we will update the main comparison in Table 1 to the following:
>
> **Test-time scaling DeepSeek-1.5B ($N=256$)**
> | Model | AIME-25 | HMMT-25 | AVG |
> | :--- | :---: | :---: | :---: |
> | VGS w/ DeepSeek-VM-1.5B (ours) | 46.7 ± 0.7 | 32.8 ± 0.8 | **39.8 ± 0.5** |
> | WMV w/ DeepSeek-VM-1.5B (ours) | 45.1 ± 2.2 | 28.9 ± 2.6 | 37.0 ± 1.7 |
> | VGS w/ DeepSeek-BT-1.5B (ours) | 40.6 ± 0.8 | 27.5 ± 0.0 | 34.1 ± 0.4 |
> | WMV w/ DeepSeek-BT-1.5B (ours) | 40.5 ± 2.9 | 24.6 ± 4.7 | 32.6 ± 2.8 |
> | VGS w/ Qwen2.5-Math-PRM-7B | 38.9 ± 1.4 | 24.2 ± 0.2 | 31.6 ± 0.7 |
> | WMV w/ Qwen2.5-Math-PRM-7B | 39.1 ± 2.1 | 24.0 ± 3.2 | 31.6 ± 1.9 |
> | VGS w/ MathShepherd-PRM-7B | 41.9 ± 1.4 | 23.9 ± 1.4 | 32.9 ± 1.0 |
> | WMV w/ MathShepherd-PRM-7B | 40.0 ± 2.5 | 25.6 ± 3.1 | 32.8 ± 2.0 |
> | MV@256 | 38.9 ± 1.9 | 24.3 ± 2.9 | 31.6 ± 1.7 |
>
> **Test-time scaling larger models with our DeepSeek-VM-1.5B:**
> | Model | AIME-25 | HMMT-25 | AVG |
> | :--- | :---: | :---: | :---: |
> | VGS w/ DeepSeek-7B ($N=128$) | 59.4 ± 0.8 | 41.1 ± 1.6 | **50.3 ± 0.9** |
> | MV w/ DeepSeek-7B ($N=128$) | 56.5 ± 1.6 | 33.8 ± 2.5 | 45.2 ± 1.5 |
>
> **Pass@1 Baselines**
> | Model | AIME-25 | HMMT-25 | AVG |
> | :--- | :---: | :---: | :---: |
> | DeepSeek-1.5B Pass@1 | 22.4 ± 4.1 | 13.0 ± 3.9 | 17.7 ± 2.8 |
> | DeepSeek-32B Pass@1 | 60.4 ± 6.0 | 42.1 ± 5.2 | 51.3 ± 4.0 |
> | Deepseek-R1 (671B) Pass@1| 70.0 ± 0.9 | 46.7 ± 2.4 | 58.4 ± 1.3 |
>
> For our camera ready, we will also update our main plots to evaluate only on AIME-25 and HMMT-25, and we will add a discussion regarding how this ensures there is no data contamination.
>
> ### 3. On the Use of Other Base Models
> We chose to focus on the DeepSeek series because they were the only open-weights reasoning models -- LLMs trained with RL to perform CoT reasoning -- at the time of submission. This focus also provided the benefit of a controlled study across different model scales (1.5B, 7B, and 14B) from the same architectural family, allowing us to investigate properties like scaling up generator sizes (Section 3.3) in a controlled manner.
>
> ### 4. On Performance in Cross-Domain Tasks
> We believe that VGS would be broadly helpful for tasks that require planning (in the RL sense), where a sequence of correct decisions is needed to reach a final goal. This includes domains like math reasoning, as explored in our paper, but could also extend to applications like creative writing, where one might guide a story's progression toward a specific ending. The primary challenge across these domains is defining a reasonable outcome-based verifier to generate the reward signal for training the value model. This is precisely why we focused our study on competition math, where the verifier is well-defined and can be reliably implemented with tools like sympy.
>
> Thank you again for your valuable questions and positive assessment. Please let us know if you have any additional questions.

---

> > ### Comment · Reviewer_YPfy · 2025-08-03
> >
> > Thank you for the thorough reply, which mostly addresses my concerns. I will maintain my high rating of the paper.

---

### Official Review · Reviewer_TuTX · 2025-07-02

**Clarity:** 4
**Significance:** 3
**Originality:** 3
**Rating:** 5
**Confidence:** 3

**Summary:**

This paper proposes a recipe for value model training; the value model is token-level and does not require defining what a step is. Using the proposed value-model-guided block-wise search, the algorithm achieves good performance compared to a few baselines on math benchmarks. Baselines include best-of-N, weighted majority voting (selecting a response from the highest-weight partition), and search algorithms that use the value model.

-	Value model is trained using a standard cross-entropy loss for classification. Input: prompt, partial response (i.e., prefix), completion starting from the prefix; reference output: label of the full response.
-	To create the dataset OpenR1-VM (2.5M reasoning responses from DeepSeek models), the authors rely on problems from OpenR1-Math. The steps involve pre-filtering, response generation (from a collection of models of different sizes; generation prefixes are determined by sampling four random prefix locations – see line 125), and post-filtering (throwing out prompts with zero accuracy from rollouts).

**Questions:**

See above

**Ethical Concerns:**

["NO or VERY MINOR ethics concerns only"]

**Final Justification:**

Tasks are limited; discussion of limitation is limited. The authors can include more baselines (e.g., other approaches based on PRMs or token-level rewards) but in general the ideas in the paper make sense intuitively.

**Limitations:**

Discussed but could expand

**Quality:**

3

**Strengths And Weaknesses:**

The method makes sense intuitively. The writing is exceptionally clear: research questions are clear in Section 3.2 for example. Figure 2 is helpful.

It’s great that the experiments are done on different base models (Qwen, Mistral) because recent work has shown that different models might have different training/fine-tuning behaviors.

I appreciate that the authors included standard deviation across three seeds in reporting the performance for math benchmarks.



---




Why does high beam width not work well? Is it due to similar issue as exposure bias in machine translation literature?

What about greedy search (width 1)?

In general I’m curious what tasks/applications would this approach be good at, what tasks/applications would this approach would this approach not make a difference or have harmful effects.

---

> ### Author Rebuttal · Authors · 2025-07-26
>
> Dear Reviewer TuTX,
>
> Thank you for the positive review and your thoughtful feedback! We are happy to clarify the points you raised and appreciate your suggestions for improving the paper.
>
> 1. On Beam Width: This is a great question. Our ablations in Figure 7 show that a beam width of 2 is consistently optimal for our method across many inference budgets. We also note that prior works with non-reasoning models [6, 32, 22] also found a low beam width of four to be optimal. We hypothesize the success of lower beam width may be due to two factors. First, it is likely easier for the value model to perform pairwise comparisons than to rank a larger set of candidate blocks. Second, given the high pass@k of DeepSeek models, it is possible that generating more diverse responses is particularly helpful for arriving the right response.
>
> 2. On Greedy Search (Width 1): We did not explicitly run a width=1 experiment given the trend that we saw in our block size ablation (Figure 6). We see a clear trend that a 4k block size was optimal across many inference budgets for our value model.
>
> 2. On the Generalizability of VGS: We believe that VGS would be broadly helpful for tasks that require planning (in the RL sense), where a sequence of correct decisions is needed to reach a final goal. This includes domains like math reasoning, as explored in our paper, but could also extend to applications such as creative writing, where one might guide a story's progression toward a specific ending. The primary challenge across these domains is defining a reasonable outcome-based verifier to generate the reward signal for training the value model. This is precisely why we focused our study on competition math, where the verifier is well-defined and can be reliably implemented with tools like sympy.
>
> Thank you again for the insightful questions and your positive evaluation. We hope our answers are helpful, and we will incorporate these points into the final version of the paper.

---

### Official Review · Reviewer_uxSh · 2025-07-05

**Clarity:** 3
**Significance:** 3
**Originality:** 2
**Rating:** 5
**Confidence:** 5

**Summary:**

This paper proposes Value-Guided Search (VGS), a method to improve the performance and computational efficiency of chain-of-thought (CoT) reasoning in large language models. The core idea is to train a token-level value model that predicts the final outcome (correct, incorrect, or incomplete) of a reasoning trace given a partial generation. This value model is trained on a large-scale dataset of 2.5 million reasoning traces, collected via a novel and scalable pipeline that does not require pre-defined "steps" or expensive human annotations. At inference time, VGS uses this value model to guide a block-wise beam search, selecting the most promising reasoning paths.

The authors demonstrate through extensive experiments on challenging competition math benchmarks (AIME and HMMT) that VGS significantly outperforms standard test-time compute (TTC) methods like majority voting and search guided by existing process reward models (PRMs). Notably, they show that a 1.5B value model can effectively guide much larger generator models (up to 14B) and reduce the FLOPs required to achieve a target accuracy. The authors are open-sourcing their dataset, model, and code.

**Questions:**

My overall assessment of the paper is positive. The following questions are intended to probe the boundaries of the method and offer suggestions for strengthening the paper even further. A thoughtful response to these points, particularly with supporting analysis where feasible, would solidify my view of this work as a top-tier contribution.

1.  **Probing the Limits of Weak-to-Strong Generalization:** The "weak-to-strong" generalization is one of the most significant claims of this work, but the data suggests it may have limits. In Table 1, the performance improvement of VGS over standard Majority Voting (MV) diminishes as the generator model gets stronger: the accuracy gain is **5.9 points** for the 7B model (56.4 vs 50.5), but drops to just **1.4 points** for the 14B model (61.0 vs 59.6). This suggests the 1.5B value model is becoming a less effective guide for the much more capable 14B model.
    *   **Question:** What is your primary hypothesis for this shrinking performance gap? Is it that the 14B model generates reasoning paths that are so complex or novel that they are fundamentally out-of-distribution (OOD) for the 1.5B value model, causing the value predictions to become less reliable?
    *   **Actionable Suggestion:** To make the paper's claims about scalability more robust, could you conduct a small-scale analysis for the rebuttal? For instance, you could analyze cases where the 14B model with VGS fails but with MV succeeds. Does the value model assign an erroneously low score to the correct path in these cases? A more ambitious but highly impactful experiment would be to train a new 1.5B value model on a small set of roll-outs from the 7B model and show if this new value model provides better guidance for the 14B generator.
    *   **Impact on Review:** A convincing analysis or a positive result from the suggested experiment would strongly reinforce the paper's central claims about the practicality and scalability of your approach, addressing what appears to be its most significant limitation.

2.  **The Critical Role of Block Size:** Your ablation on search block size (Figure 6) shows that a large block of 4096 tokens is consistently optimal. This is a fascinating and perhaps counter-intuitive result, as one might expect that more frequent guidance (i.e., smaller blocks) would be beneficial for correcting errors early.
    *   **Question:** Why do you believe such a large block size is optimal? I see three competing hypotheses: (a) **Signal-to-Noise:** The value model requires a large context to make an accurate prediction, and its judgments on shorter blocks are too noisy to be useful guides. (b) **Task-Specific Coherence:** The reasoning steps in competition math problems are long and logically self-contained, so interrupting the generator mid-"thought" is detrimental. (c) **Amortized Cost:** The overhead of invoking the value model is non-trivial, and using fewer, larger blocks is simply more computationally efficient for a given budget. Could you provide your perspective on these or offer an alternative hypothesis?
    *   **Actionable Suggestion:** This is a key parameter of your method. To help the reader understand this result, could you provide a qualitative example in the appendix showing a generation from a small-block-size search (e.g., 256) that fails, contrasted with a successful 4096-block-size search on the same problem? A clear discussion of these trade-offs in Section 4.1 would significantly improve the paper.

3.  **From Binary Math Problems to Nuanced Domains:** The success of your data collection pipeline and value model training hinges on the availability of an unambiguous, binary correctness signal from `math-verify`. This is an ideal scenario that is not representative of many other complex reasoning tasks.
    *   **Question:** How do you envision the VGS framework being adapted to domains with more nuanced or non-binary rewards? Consider code generation, where a program might pass some unit tests but not others. Would you adapt the value model to predict a continuous score (e.g., fraction of tests passed), or would you define a more complex set of discrete outcome classes (e.g., 'passes all tests', 'fails compilation', 'fails runtime tests', 'infinite loop', 'incomplete')? How would the learning objective and the final WMV aggregation handle such a signal?
    *   **Actionable Suggestion:** While a full experimental evaluation is outside the scope of the rebuttal, a more detailed discussion of these necessary adaptations would greatly strengthen the paper by clarifying the boundaries and future trajectory of this line of research. I suggest adding a dedicated paragraph on this topic in your "Limitations" or "Conclusion" section. A thoughtful response here would demonstrate a deeper consideration of the method's generalizability.

4.  **The Power of Final Aggregation via WMV:** The finding that using Weighted Majority Vote (WMV) at the end of the search is crucial, and significantly outperforms Best-of-N (BoN) (Figure 3), is a very important and subtle result. It implies that even after guiding the generation block-by-block, the final value scores across the surviving beams still contain useful, independent information. In a perfect world with a perfect value model and search, the single highest-scoring path should be the best.
    *   **Question:** What is your explanation for why WMV is so much more effective? Does this suggest that the value model's scores are inherently noisy, and that a single high score on one beam might be an outlier, making aggregation across multiple diverse beams a more robust ensembling strategy?
    *   **Actionable Suggestion:** This is a key insight into the practical use of process-level scorers. Could you provide a brief analysis to support your hypothesis? For example, you could measure the variance of value scores across the final beams, comparing cases where BoN fails and WMV succeeds. A high variance in these cases would support the "noisy value model" hypothesis. Highlighting this finding and its implications more explicitly in the main text would be a valuable addition.

**Ethical Concerns:**

["NO or VERY MINOR ethics concerns only"]

**Final Justification:**

I thank the authors for an exemplary rebuttal that was both thorough and directly responsive to my questions. They have convincingly resolved all of my initial reservations, significantly strengthening my confidence in this work's contribution.

The key issues that have now been addressed are:

1.  **Limits of Generalization:** My concerns about the diminishing performance gains ("weak-to-strong" generalization) have been fully addressed. The authors confirmed the out-of-distribution (OOD) challenge and importantly clarified the practical, intended use case (guiding with a similarly-sized value model), which resolves the apparent scaling limitation.
2.  **Optimal Block Size:** The authors provided a compelling explanation for why a large block size is optimal, combining the need for a clear signal-to-noise ratio with the importance of coherent reasoning. Their commitment to add the suggested qualitative example to the appendix will make this core design choice much clearer to readers.
3.  **Adaptation to Nuanced Domains:** The thoughtful discussion on adapting the VGS framework to more nuanced domains (e.g., using distributional RL) has effectively addressed my questions about the method's generalizability and clarifies the future trajectory of this research.
4.  **The Power of WMV Aggregation:** The explanation for why Weighted Majority Vote (WMV) outperforms Best-of-N—and the commitment to add a new ablation to support this claim—provides a critical and novel insight into the practical application of value-based search methods.

In summary, the authors' comprehensive and actionable responses have turned a solid paper into an even stronger one. The promised additions will substantially improve the final version. Therefore, I have raised my score from 4 (Borderline Accept) to **5 (Accept)** and now strongly advocate for this paper's acceptance.

**Limitations:**

The authors have a dedicated limitations section that appropriately identifies the main limitation: the value model is trained on roll-outs from a specific policy and will need to be retrained to maintain optimal performance as generator models advance.

Two areas for improvement would be:
1.  **Elaborating on "Verifiable Domains":** The authors state their pipeline can be adapted to "similar verifiable domains." It would be helpful to expand on what constitutes a "verifiable domain" beyond binary pass/fail and discuss the challenges for tasks with more nuanced, multi-faceted, or human-in-the-loop reward structures.
2.  **Potential Negative Societal Impact:** While the risk for math problem solving is low, the paper contributes to the broader area of improving automated reasoning. A brief mention of the potential dual-use nature of such technologies (e.g., generating highly convincing but subtly flawed arguments at scale) would be appropriate for a paper at this level.

**Paper Formatting Concerns:**

No major formatting concerns.

**Quality:**

4

**Strengths And Weaknesses:**

This is a well-executed paper that makes a significant contribution to the important problem of scaling LLM reasoning.

**Strengths:**

1.  **Significance and Impact:** The paper tackles a critical challenge in the LLM landscape: the immense computational cost of generating long, high-quality reasoning chains. The proposed VGS method offers a practical and effective solution that demonstrably improves the trade-off between performance and compute. The finding that a small value model can effectively guide a much larger generator ("weak-to-strong" generalization) is particularly significant, as it suggests a cheaper and more scalable path to improving frontier models.

2.  **Technical Quality and Rigor:** The experimental evaluation is exemplary.
    *   **Challenging Benchmarks:** The use of recent and difficult benchmarks (AIME/HMMT 2024 & 2025) provides a convincing testbed for their method, moving beyond potentially saturated benchmarks.
    *   **Thorough Ablations:** The authors conduct a comprehensive set of ablations that validate their key design choices. The analyses of block size (Fig. 6), beam width (Fig. 7), the benefit of DVTS, and the direct comparison between value-guided and random search (Fig. 8) provide strong support for their methodology.
    *   **Strong Baselines:** The comparison against not just standard TTC methods but also state-of-the-art 7B PRM models strengthens the paper's claims. The fact that VGS outperforms these larger, more complex PRMs highlights the effectiveness of the authors' proposed value model training scheme.
    *   **Novel Empirical Finding:** The discovery that using Weighted Majority Vote (WMV) for final aggregation is critical to the success of VGS is a subtle but important finding that is novel to this work.

3.  **Originality and Novelty:** The primary contribution is the novel and scalable pipeline for training a token-level value model without relying on a pre-defined notion of a "step." This elegantly sidesteps a major limitation of prior work on Process Reward Models (PRMs), making process-level guidance practical for long-context models like DeepSeek. While the components (beam search, value models) are known, their combination into the block-wise VGS framework, the specific step-free data generation strategy, and the successful application to long-form math reasoning is a novel and impactful contribution.

4.  **Clarity and Presentation:** The paper is exceptionally well-written, clear, and easy to follow. The problem statement is well-motivated, and the proposed method is explained logically. Figure 2 provides an excellent, intuitive summary of the entire data collection and search pipeline. The distinction from and advantages over prior work are clearly articulated.

5.  **Reproducibility:** The commitment to open-sourcing the dataset (OpenR1-VM), the trained 1.5B value model, and the codebase is a major strength and a great service to the research community.

**Weaknesses:**

1.  **Limited Generality of the Value Model:** The value model is trained on roll-outs from a single, fixed policy (DeepSeek-R1-Distill-1.5B). While the weak-to-strong results are promising, Table 1 shows that the performance gap between VGS and standard Majority Voting (MV) narrows as the generator model size increases from 7B to 14B. This suggests the 1.5B value model may be approaching the limits of its ability to effectively guide a much stronger generator. The paper would be strengthened by a deeper discussion of this OOD issue and the expected scaling limits.

2.  **Domain Specificity:** The entire evaluation is conducted on competition math problems. This domain benefits from having an unambiguous, binary correctness signal that can be checked automatically (`math-verify`). It is unclear how the VGS pipeline, particularly the 3-class (correct, incorrect, incomplete) data labeling, would translate to more open-ended domains where rewards are nuanced or require human evaluation (e.g., creative writing, summarization quality, or complex code generation).

3.  **Fixed Search Hyperparameters:** The authors use a fixed block size and beam width for all problems, which they frame as a practical advantage over methods that require per-problem tuning. However, this "one-size-fits-all" approach might be suboptimal. It's possible that different problems could benefit from different search strategies (e.g., a wider search for harder problems), and a discussion of this trade-off between practicality and potentially higher performance would be welcome.

---

> ### Author Rebuttal · Authors · 2025-07-27
>
> Dear Reviewer uxSh,
>
> Thank you for your exceptionally thorough and insightful review. We are grateful for your detailed positive feedback on our work's significance, rigor, and clarity. We really appreciate the opportunity to respond to your questions and hopefully solidify your view of VGS as a top-tier contribution.
>
> # Responses to Questions:
> 1. On the Limits of Weak-to-Strong Generalization: Your hypothesis is correct. The shrinking performance gap is likely an out-of-distribution (OOD) issue, as the 14B generator produces reasoning paths that are qualitatively different from the 1.5B rollouts our value model was trained on.
>
>     While retraining the value model on rollouts from a stronger generator is the ideal experiment, it is unfortunately too computationally expensive for us to perform during the rebuttal period. We also want to clarify that demonstrating weak-to-strong generalization, while an interesting result, is not the main intended use case of VGS. When applying VGS to guide frontier models, we envision training a value function of a similar size to the generator policy itself. Since the value model is only queried once per block, the computational overhead of this approach is negligible, as we show in Appendix H.
>
> 2. On the Critical Role of Block Size (4096 tokens): We agree this is a fascinating result, and we believe the optimality of a large block size is driven by a combination of two of your hypotheses:
>     (a) Signal-to-Noise: The value model requires sufficient context to make an accurate prediction. A short block may not contain enough information to distinguish a promising path from a dead end, leading to noisy and unreliable guidance.
>     (b) Task-Specific Coherence: Competition math problems often require long, uninterrupted lines of reasoning. We found that interrupting the generator too frequently with smaller blocks is disruptive to this coherent thought process.
>     We don't think the amortized cost hypothesis (c) is a reason because the performance also degrades for blocks larger than 4096, which would not be explained by efficiency alone. We will add a qualitative example to the appendix, as you suggested, to illustrate this trade-off and expand this discussion in Section 4.1.
>
> 3. On Adapting VGS to Nuanced Domains: This is a critical question. The notion of a value function is very general and can be applied to any type of reward signal. For domains with more nuanced rewards, we would adapt the value model's training as follows:
>     * For continuous rewards (e.g., the fraction of unit tests passed), we would suggest moving beyond simple regression and using a distributional RL (Bellemare et al. 2017) approach to model the entire distribution of possible rewards. A distributional approach to value-based RL has robustly shown improved downstream decision making and is supported by a large body of prior works (Imani et al. 2024, Farebrother et al. 2024, Wang et al. 2025).
>     * For multi-faceted discrete rewards, we would use a more complex set of outcome classes, as you suggest.
>     In general, we believe that VGS would be broadly helpful for tasks that require planning (in the RL sense), where a sequence of correct decisions is needed to reach a final goal. This includes domains like math reasoning, as explored in our paper, but could also extend to applications like creative writing, where one might guide a story's progression toward a specific ending. The primary challenge across these domains is defining a reasonable outcome-based verifier to generate the reward signal for training the value model. This is precisely why we focused our study on competition math, where the verifier is well-defined and can be reliably implemented with tools like `sympy`.
>
> 4. On the Power of Final Aggregation via WMV: This is a key insight. Our explanation for why Weighted Majority Vote (WMV) is more effective than Best-of-N (BoN) is twofold:
>     * First, WMV is more scalable against the value model being "hacked" at large inference budgets. With many samples, BoN is more likely to select an outlier—a flawed path that receives an erroneously high score. WMV, by relying on consensus, is more robust to such outliers.
>     * Second, WMV provides an implicit filter for unfinished responses. Incomplete generations will not produce the same final answer and thus will not be part of a majority cluster. BoN does not have this filter, and our value model sometimes assigns an unduly high score to an incomplete trace.
>     As a concrete action, we will add results to the final version showing the performance of BoN after manually filtering for unfinished responses to further support this analysis.
>
> Thank you again for your detailed and expert feedback. We believe that incorporating these analyses and discussions will significantly strengthen our paper. We are grateful for your review.
>
> --------------------
> Citations
>
> [1] Bellemare et al. 2017: "A distributional perspective on reinforcement learning", ICML 2017.
>
> [2] Imani et al. 2024: "Investigating the histogram loss in regression", arXiv 2024
>
> [3] Farebrother et al. 2024: "Stop regressing: Training value functions via classification for scalable deep rl", ICML 2024
>
> [4] Wang et al. 2025: "The central role of the loss function in reinforcement learning". Statistical Science, 2025

---

### Official Review · Reviewer_4pBh · 2025-07-08

**Clarity:** 3
**Significance:** 4
**Originality:** 2
**Rating:** 5
**Confidence:** 3

**Summary:**

Based on a newly collected (and also released) large-scale dataset with reasoning paths a value model is learned for guiding DeepSeek LLMs on reasoning tasks. The value model is trained on partial responses and completions based on a more cost-efficient, distilled  reference LLM and can be combined with multiple blockwise, chain-of-thought reasoning strategies such as best-first search or beam search. The resulting value-model guided search achieves good performance and efficiency on standard math benchmarks like AIME or HMMT.

**Questions:**

* Regarding the Experiment in Section 4.2: If I understand it correctly, the blocks are selected uniformly at random. Selecting the blocks according to the likelihood under the LLM seems like a stronger -- and equally expensive and simple -- baseline to me.  Do I misunderstand the experiment or is there another reason for why you didn't use this ablation?
* How does the method conceptually compare to tokenwise reward-guided text generation?

  > **A Critical Look At Tokenwise Reward-Guided Text Generation**, Rashid et al., 2025, https://arxiv.org/pdf/2406.07780

   The training on partial sequences as describe in Section 2.1 strongly reminds me of the method in the above reference, but the related work section does not discuss it.

* I assume tree-of-thought based reasoning is in an entirely different league of computational budget than chain-of-thought based methods. Is that why you did not empirically compare the value model with tree-of-thought based approaches or are there other reasons?
* Can the value model also be applied to tree-of-thought based reasoning?

**Ethical Concerns:**

["NO or VERY MINOR ethics concerns only"]

**Final Justification:**

The authors rebuttal mostly addressed my concerns. I also read the other reviews and I agree with Reviewer oEdM, that an empirical comparison with step-level or token-level reward guided models would be interesting. On the other hand, if there is already some evidence in prior work that block-wise guidance performs better than token-level guidance (e.g. the paper of Mudgal et al. 2024, which I was not familiar with), I am willing to accept that such a comparison is not absolutely necessary. I will therefore maintain my positive score.

**Limitations:**

yes

**Paper Formatting Concerns:**

no formatting concerns

**Quality:**

3

**Strengths And Weaknesses:**

**Strengths**:
* significant performance and efficiency gains on standard reasoning benchmarks
* robustness towards hyperparameter choice across a  wide range of computational budgets
* methods scales well to long contexts
* release of large-scale dataset with reasoning-responses that might also be of use for future work

**Weaknesses**:
* The empirical evaluation exclusively focuses on LLMs from the DeepSeek family
* Some parts of the paper are not fully clear to me. I list my questions below.

**Minor comment**:
* I think it would further enhance the readability of Table 1, if the best and 2nd best performance on each of the benchmarks would also be highlighted, and not just the average performance.

---

> ### Author Rebuttal · Authors · 2025-07-26
>
> Dear Reviewer 4pBh,
>
> Thank you for your positive and insightful review! We are happy to clarify the points you raised and appreciate your suggestions for improving the paper.
>
> ### On the Scope of Evaluation:
> We chose to focus on the DeepSeek series because they were the only open-weights reasoning models -- LLMs trained with RL to perform CoT reasoning -- at the time of submission. This focus also provided the benefit of a controlled study across different model scales (1.5B, 7B, and 14B) from the same architectural family, allowing us to investigate properties like scaling up generator sizes (Section 3.3) in a controlled manner.
>
> ### On the Minor Comment:
> Thanks for this suggestion. We will update the table to highlight the best and second-best performing methods on each individual benchmark in the camera-ready version.
>
> ### Responses to Questions:
> 1. Regarding the Random Search Ablation (Section 4.2): Your understanding is correct. Our intent is to cleanly isolate the effect of using our value model for selecting intermediate blocks versus having no selection rule at all. By removing the value signal entirely, we can directly measure its contribution to the search process. We agree that a likelihood-based selection is a strong baseline for a verifier-free search, and we will run this comparison and add a discussion to the final version.
>
> 2. Comparison to (Rashid et al., 2025): Thank you for this reference; the work is certainly related and merits a citation and discussion. There are two primary issues / challenges with applying reward guidance at the token level that make it less efficient and scalable:
>     * Computational Cost: Token-wise guidance requires a classifier forward pass for every generated token [1,2,3], which is substantially more expensive than our block-wise method that only queries the value model once per block (e.g., every 4096 tokens).
>     * Compounding Errors: An imperfect reward or value model, when queried too frequently, can introduce cascading errors that derail the generation process. Indeed, prior work by Mudgal et al. 2024 compared block-wise versus token-wise guidance and found that block-wise approaches performed better, which informed our decision to focus on these methods.
>
>     We will expand on this discussion and add a citation to Rashid et al. to our related work section.
> 3. Comparison to Tree-of-Thought (ToT) Approaches: Thanks for the great question. Our Value-Guided Search can be viewed as an instantiation of the broader ToT framework, where our value model serves as the verifier to score different reasoning paths. Specifically, in the ToT framework, our policy model $\pi$ corresponds to the "thought generator", and our value model corresponds to the "state evaluator", which scores the generated blocks to determine which reasoning paths are most promising to pursue. Thus, VGS can be viewed as a specific, highly effective instantiation of this general framework. We show this particular approach is an efficient and powerful method for improving long-context reasoning, but we agree that exploring more complex ToT search strategies with a value model evaluator can lead to even more efficient reasoning, which we believe is a promising direction for future work.
>
> Thank you again for your supportive review and constructive feedback. We will incorporate these clarifications and additions into our camera-ready version and believe your suggestions will significantly strengthen the paper.
>
> ---------------
> Citations:
>
> [1] Rashid, Ahmad, et al. "A critical look at tokenwise reward-guided text generation." arXiv preprint arXiv:2406.07780 (2024).
>
> [2] Mudgal, Sidharth, et al. "Controlled decoding from language models." arXiv preprint arXiv:2310.17022 (2023).
>
> [3] Zhou, Jin Peng, et al. "$ Q\sharp $: Provably Optimal Distributional RL for LLM Post-Training." arXiv preprint arXiv:2502.20548 (2025).

---

> > ### Comment · Reviewer_4pBh · 2025-08-05
> >
> > Thank you for the clarifications.
> >
> > I also read the other reviews and I agree with Reviewer oEdM, that an empirical comparison with step-level or token-level reward guided models would be interesting. On the other hand, if there is already some evidence in prior work that block-wise guidance performs better than token-level guidance (e.g. the paper of Mudgal et al. 2024, which I was not familiar with), I am willing to accept that such a comparison is not absolutely necessary. I will therefore maintain my positive score.

---

### Official Review · Reviewer_oEdM · 2025-07-09

**Clarity:** 2
**Significance:** 2
**Originality:** 4
**Rating:** 2
**Confidence:** 4

**Summary:**

This paper proposes Value-Guided Search (VGS), a method to improve the efficiency and effectiveness of long-context chain-of-thought (CoT) reasoning in language models. The central idea is to train a token-level value model that predicts the outcome of a reasoning trace without requiring a fine-grained definition of “step.” This value model is then used to guide block-wise search (e.g., beam search with blocks of 4096 tokens), enabling more effective inference-time compute scaling than standard test-time compute (TTC) methods such as majority voting or best-of-N. The authors build a large dataset of 2.5M math reasoning traces from DeepSeek models and demonstrate strong results on four high-school math benchmarks (AIME & HMMT, 2024/2025), showing that their method is competitive with or better than step-level PRM-based baselines. All code, models, and data are promised to be open-sourced.

**Questions:**

See weaknesses.

**Ethical Concerns:**

["NO or VERY MINOR ethics concerns only"]

**Final Justification:**

I have read the author's rebuttal and the comments from other reviewers, and after a comprehensive evaluation, I have decided to maintain my original score.

**Limitations:**

Yes

**Quality:**

3

**Strengths And Weaknesses:**

Strengths:

* Clear and focused motivation: The paper is framed around the growing cost and inefficiency of long-context reasoning in LLMs, a timely and important topic.

* Strong empirical results: The proposed method consistently outperforms standard TTC baselines (e.g., MV, WMV, BoN) and state-of-the-art PRMs across multiple competition math datasets, even when guiding larger generators like DeepSeek-14B.

* Ablation and sensitivity studies are comprehensive: The authors evaluate block size, beam width, DVTS parallelism, and even hybrid/random guidance settings, increasing the paper’s technical completeness.

Weaknesses:

* High training cost and limited generalization of value model: (1) Although the authors claim their training pipeline is scalable, their value model (trained on DeepSeek-1.5B rollouts) requires 96×H100 GPUs running for 24 hours, not including tuning iterations. This is non-trivial for most researchers. (2) The trained value model shows signs of out-of-distribution (OOD) degradation when guiding larger generators (e.g., DeepSeek-14B), which indicates that a separate value model may be needed for each generator model. This introduces a significant maintenance and compute burden, limiting real-world applicability despite the authors' claim (Line 217–218) that this is not a “practical concern.”

* Unclear advantage over step-level supervision: (1) A key motivation of the paper is that step-level supervision is difficult to define and expensive to collect. However, the authors do not quantitatively compare the cost or quality of their token-level labeling approach to step-level PRMs. (2) In practice, step definitions via rule-based chunking (e.g., newline delimiters) have been shown to work well in PRM training. Therefore, it remains unclear whether the additional complexity of collecting roll-in/roll-out pairs and full-trace supervision is actually more cost-effective than existing PRM alternatives.

---

> ### Author Rebuttal · Authors · 2025-07-26
>
> Dear Reviewer oEdM,
>
> Thank you for your constructive review. We really appreciate the opportunity to respond to your questions and concerns.
>
> ### Regarding Value Model Training Cost and Generalization:
>
> **(1) On Training Cost & Scalability**: We apologize for the lack of clarity -- by "scalable," we refer specifically to our data collection pipeline. Our method avoids the need for expensive, step-by-step human or LLM-as-a-Judge annotations required by prior PRM approaches. This simplification is what enabled us to collect a large-scale dataset of 2.5 million reasoning traces (each having up to 16k tokens). Training on a dataset of this magnitude indeed requires significant compute. To ensure this effort benefits the entire community, we are open-sourcing the full dataset, model checkpoints, and codebase, allowing other researchers to bypass this one-time cost.
>
> **(2) On Generalization**: Thank you for this point. While it is true that our 1.5B value model shows signs of OOD degradation when guiding DeepSeek-14B, we want to highlight that our value model still strongly improves MV when guiding DeepSeek-7B. We view this as a successful demonstration of weak-to-strong generalization: our value model can guide a ~5x larger model to strongly improve the MV baseline (Table 1, VGS guiding DeepSeek-7B has 56.4% accuracy vs. MV with DeepSeek-7B has 50.5% accuracy, so VGS shows a 6% improvement).
>
> Moreover, in practice, value guidance is only a small fraction of the total inference compute as we showed in Appendix H, since the value model is only queried at every block of tokens. Thus, we believe training a large value model (of the same size as the generator or even larger) is a reasonable investment and we believe VGS is a promising way to for scaling test-time compute of reasoning models.
>
> ### Regarding the Advantage over Step-Level Supervision:
>
> We agree it is difficult to provide a direct "apples-to-apples" cost comparison. The key advantage of our approach is that it mitigates the need to define a "step," a notion that is difficult to formalize for complex reasoning traces (Guo et al. 2025). For example, if one were to use a simple heuristic like newlines, a single reasoning trace from DeepSeek can easily exceed 100 "steps." Using an LLM-as-a-judge for our 2.5M trace dataset would have required over 250 million such judgments, a staggering and impractical cost. In contrast, our outcome-based method requires only one cheap verification per reasoning trace.
>
> We note that our approach is philosophically aligned with Sutton's "bitter lesson," favoring general methods that scale with data and compute over those relying on human-defined structures such as a "step." Indeed, our VGS method using a 1.5B value model, empirically outperforms a 7B PRM baseline trained with step-level feedback (Table 1, VGS w/ DeepSeek-VM-1.5B has 45.7% accuracy vs. VGS w/ Qwen2.5-Math-PRM-7B has 40.2% accuracy). Ultimately, we hypothesize it may be useful to learn an appropriate notion of "step" beyond the newline heuristic, and we leave that as promising future work.
>
> We hope these clarifications address your concerns and highlight the significance of our contributions. Thank you again for your time and valuable feedback.

---

> ### Comment · Reviewer_oEdM · 2025-08-08
>
> I have read the rebuttal and will update the score accordingly.

---

### Decision · Program_Chairs · 2025-09-17

**Decision:**

Accept (poster)

**Comment:**

(a) Summarize the scientific claims and findings of the paper based on your own reading and characterizations from the reviewers.

This paper introduces Value-Guided Search (VGS), a method for improving the efficiency and accuracy of long-context, chain-of-thought (CoT) reasoning in language models. The central contribution is a scalable pipeline for training a token-level value model that predicts the outcome of a reasoning trace without relying on a pre-defined, fine-grained notion of a "step," which has been a limitation of prior Process Reward Models (PRMs). The authors collected a large-scale dataset of 2.5 million reasoning traces to train a 1.5B parameter value model. At inference time, this model guides a block-wise beam search, and a final weighted majority vote (WMV) is used for aggregation.

The key findings are that VGS significantly outperforms standard test-time compute methods like majority voting and best-of-N sampling on challenging competition math benchmarks (AIME and HMMT). The method is shown to be more computationally efficient, reducing the FLOPs required to reach a target performance level. Furthermore, the authors demonstrate a "weak-to-strong" generalization capability, where their 1.5B value model successfully improves the performance of a larger 7B generator model. The authors commit to open-sourcing their dataset, model, and codebase.

(b) What are the strengths of the paper?

The paper's strengths were highlighted by a majority of the reviewers:

Significance and Timeliness: It addresses the critical and timely problem of improving the computational efficiency of long-context reasoning in LLMs.

Novel Methodology: The "step-free" approach to training the value model is an elegant and practical innovation that sidesteps a key difficulty in creating process-level supervision for complex reasoning tasks.

Strong Empirical Results: The method achieves substantial performance gains on difficult and recent benchmarks. The evaluation is rigorous, including comprehensive ablations on key design choices (e.g., block size, beam width) and comparisons to strong baselines, including existing PRMs.

Clarity and Reproducibility: The paper is well-written and easy to follow. The commitment to open-sourcing the large-scale dataset, the trained value model, and the implementation is a significant contribution to the community.

(c) What are the weaknesses of the paper? What might be missing in the submission?

The reviewers identified a few weaknesses, which were discussed at length during the rebuttal period:

Generalization of the Value Model: The value model is trained on data from a specific model family (DeepSeek) and shows diminishing returns when guiding a much larger, out-of-distribution generator (DeepSeek-14B). This suggests that the value model may need to be retrained or scaled up as generator models advance, which introduces a maintenance and compute burden.

Domain Specificity: The evaluation is confined to the domain of competition mathematics, which benefits from a clear, automatically verifiable correctness signal. How the data collection and training pipeline would adapt to more open-ended domains with nuanced rewards (e.g., summarization, creative writing) is not fully explored.

High Training Cost: While the authors emphasize the scalability of data collection, the one-time training cost for the value model is substantial (requiring 96 H100 GPUs for 24 hours), which could be a barrier for academic labs. However, this is mitigated by the open-sourcing of the final model.

(d) Provide the most important reasons for your decision to accept/reject.

The decision to accept is based on the paper's novel and practical solution to an important problem, backed by strong and thorough empirical evidence. The "step-free" value modeling approach is a significant conceptual contribution that advances the state of the art beyond existing PRMs. The demonstrated improvements in both performance and computational efficiency are substantial. While the weaknesses regarding generalization and domain specificity are valid, they represent limitations and avenues for future work rather than fatal flaws. The authors' commitment to releasing all artifacts further strengthens the paper's value and potential impact on the field. The overall positive assessment from four out of five reviewers, with one becoming a strong advocate after the rebuttal, confirms the paper's quality and readiness for publication.

(e) Summarize the discussion and changes during the rebuttal period. What were the points raised by the reviewers? How were each of these points addressed by the authors? How did you weigh in each point in your final decision?

The discussion period was highly productive and significantly clarified the paper's contributions and limitations.

Reviewer oEdM raised the primary dissenting points regarding the high training cost and the limited generalization of the value model to OOD generators. The authors clarified that their claim of "scalability" referred to data collection (avoiding expensive step-wise annotation) and acknowledged the OOD issue as a known trade-off. They argued that in practice, one would train a value model of a similar size to the generator, making the inference overhead small. While this reviewer was not fully convinced and maintained their score, their points were acknowledged as valid limitations of the approach.

Reviewer uxSh provided an exceptionally detailed and positive review, asking deep questions about the limits of generalization, the reason for the large optimal block size, adaptation to nuanced domains, and the effectiveness of the final WMV aggregation. The authors' rebuttal was exemplary, providing compelling hypotheses and promising additional analyses and qualitative examples in the final version. This exchange turned a solid paper into an even stronger one, leading the reviewer to raise their score and become a strong advocate for acceptance.

Reviewers 4pBh, TuTX, and YPfy raised other important points, including comparisons to related work (token-wise guidance, ToT), the choice of hyperparameters, data contamination, and generalizability. The authors addressed each of these points convincingly. They contextualized VGS within the ToT framework, differentiated it from more costly token-wise methods, proposed a clear strategy to control for data contamination, and discussed the challenges and potential solutions for applying VGS to other domains. These reviewers were all satisfied with the responses and maintained their positive scores.

In my final decision, I weighed the strong support from the majority of reviewers against the concerns of one. The authors' rebuttal was thorough and effective in addressing most concerns, particularly those from reviewer uxSh, which added depth to the paper's contributions. The concerns raised by reviewer oEdM are valid limitations but do not undermine the novelty and empirical strength of the work. The consensus among the other four reviewers, combined with the quality of the paper and the rebuttal, strongly supports acceptance, which is my recommendation.